# Reduced efficacy of HIV-1 integrase inhibitors in patients with drug resistance mutations in reverse transcriptase

Mark J. Siedner [1,2,3,4✉], Michelle A. Moorhouse [5], Bryony Simmons [6], Tulio de Oliveira[4,7], Richard Lessells [4,7], Jennifer Giandhari[4,7], Stephen A. Kemp[8], Benjamin Chimukangara[4,7,9], Godspower Akpomiemie[5], Celicia M. Serenata [5], Willem D. F. Venter[5], Andrew Hill[10,11] & Ravindra K. Gupta [1,4,8,11✉]

Little is known about the impact of pretreatment drug resistance (PDR) on the efficacy of second generation integrase inhibitors. We sequenced pretreatment plasma specimens from the ADVANCE trial (NCT03122262). Our primary outcome was 96-week virologic success, defined as a sustained viral load <1000 copies/mL from 12 weeks onwards, <200 copies/mL from 24 weeks onwards, and <50 copies/mL after 48 weeks. Here we report how this outcome was impacted by PDR, defined by the World Health Organization (WHO) mutation list. Of 1053 trial participants, 874 (83%) have successful sequencing, including 289 (33%) randomized to EFV-based therapy and 585 (67%) randomized to DTG-based therapy. Fourteen percent (122/874) have ≥1 WHO-defined mutation, of which 98% (120/122) are NNRTI mutations. Rates of virologic suppression are lower in the total cohort among those with PDR 65% (73/112) compared to those without PDR (85% [605/713], $P < 0.001$), and for those on EFV-based treatment (60% [12/20] vs 86% [214/248], $P = 0.002$) and for those on DTG-based treatment (61/92 [66%] vs 84% [391/465] $P < 0.001$, $P$ for interaction by regimen 0.49). Results are similar in multivariable models adjusted for clinical characteristics and adherence. NNRTI resistance prior to treatment is associated with long-term failure of integrase inhibitor-containing first-line regimens, and portends high rates of first-line failure in sub Saharan Africa.

[1] Africa Health Research Institute, KwaZulu-Natal, South Africa. [2] Massachusetts General Hospital, Boston, MA, USA. [3] Harvard Medical School, Boston, MA, USA. [4] University of KwaZulu-Natal, Durban, South Africa. [5] Ezintsha, Wits Reproductive Health and HIV Institute, University of Witwatersrand, Johannesburg, South Africa. [6] Department of Infectious Disease, Imperial College, London, UK. [7] KwaZulu-Natal Research Innovation and Sequencing Platform, Durban, South Africa. [8] University of Cambridge, Cambridge, UK. [9] Centre for the AIDS Programme of Research in South Africa (CAPRISA), Durban, South Africa. [10] University of Liverpool, Liverpool, UK. [11] These authors contributed equally: Andrew Hill, Ravindra K. Gupta. ✉email: Mark.siedner@ahri.org; Rkg20@cam.ac.uk

The increasing prevalence of non-nucleoside reverse transcriptase inhibitor (NNRTI) resistance in those initiating or re-initiating antiretroviral therapy (ART)[1], along with the advantageous safety, potency, and cost-effectiveness characteristics of dolutegravir (DTG)[2], prompted the World Health Organization (WHO) to recommend DTG-based ART as a preferred first-line regimen[3]. However, recent concerns about DTG have emerged. For example, early data suggested a slight increased risk of neural tube defects following DTG exposure in pregnancy, although more recent data have been reassuring[4]. Secondly, greater weight gain was observed in patients treated with DTG compared to efavirenz (EFV) in two clinical trials in sub-Saharan Africa, as well as in cohort studies from the North America and Europe, leading to concerns about long-term effects of obesity with lifelong ART[5–9].

Whilst cost-effectiveness analyses continue to support the use of DTG as first-line therapy despite these issues[10], the WHO and others are revisiting targeted use of efavirenz (EFV). Concerns remain about the use of EFV with widespread NNRTI resistance, which exceeds 10−15% in much of sub-Saharan Africa[11]. Pretreatment NNRTI resistance has been associated with a 2−3-fold greater risk of virologic failure (VF) for people initiating NNRTI-based regimens, both with older combinations such as nevirapine (NVP) and with EFV[12–14]. By contrast, the ANRS 12249 Treatment as Prevention Trial[15] reported that the most common NNRTI mutation, K103N, when detected alone, was not associated with increased risk of VF on an NNRTI-based single tablet regimen containing tenofovir, emtricitabine and efavirenz[16]. A study in Kenya similarly suggested isolated K103N might have limited impact on EFV-based ART[17].

These conflicting data have generated controversy in the field on optimal first-line regimens to balance safety, tolerability, cost, and the impact of circulating NNRTI drug resistance on virologic outcomes. Although clinical trial data in the United States suggest that DTG performs exceptionally well in ART-naive individuals and as a switch regimen in the absence of significant background resistance[18–20], there is relatively little data available on the efficacy of DTG in the context of high circulating NNRTI resistance. In the DAWNING trial, in which individuals failing NNRTIs were randomized to DTG or lopinavir/ritonavir, and over 90% had some evidence of NNRTI resistance, approximately 84% of individuals in the DTG arm achieved virologic suppression at 48 weeks. Notably the proportion of people suppressed at 48 weeks on DTG arm was lower than in most prior clinical trials, albeit of first-line therapy[21]. As such, additional studies are needed to better elucidate the impact of pretreatment NNRTI drug resistance on virologic outcomes with both EFV-based and DTG-based used first-line regimens in the region.

Here we report results of next-generation sequencing of stored plasma specimens from participants in the ADVANCE clinical trial to determine the contributions of NNRTI pretreatment drug resistance (PDR) on 96-week virologic outcomes for individuals initiating EFV- and DTG-based ART. We hypothesize that NNRTI PDR significantly affects efficacy of EFV-containing regimens but has a negligible effect on outcomes for those initiating DTG-based therapy.

## Results

**Study population.** A total of 1053 individuals were enrolled in the ADVANCE trial. Of these, 991 (94%) consented for specimen storage and had pretreatment plasma available for testing, and 874 (83%) had successful sequencing of a pretreatment plasma specimen (Fig. 1). We found no differences in clinical or demographic characteristics between those who successfully underwent sequencing and those who did not (Supplementary information).

Of participants included in PDR analyses, 289 (33%) were randomized to an EFV-based regimen and 585 (67%) were randomized to a DTG-based regimen. At the time of data extraction, all had completed observation up to 96 weeks. A total of 48 and 82 individuals were excluded from our primary and secondary analyses, respectively, for not remaining in the study to 12 or 24 weeks. There were no differences by treatment regimen in clinical or demographic factors in our primary analytic sample (Table 1). However, individuals starting DTG-based regimens had a higher prevalence of PDR than those initiating EFV-based regimens (16.5 vs 7.4%, P < 0.001).

**Pretreatment drug resistance.** Approximately 14% (122/874) of individuals had at least one WHO-defined PDR mutation at variant frequencies of 20% or greater (Fig. 2). The majority of PDR was accounted for by mutations conferring resistance to NNRTIs, with over 98% (120/122) of those harboring WHO-defined PDR having at least one NNRTI mutation. The most common single mutation was K103N, present in 9% (81/874). Only 20 (2%) individuals had a nucleoside reverse transcriptase inhibitor (NRTI) mutation, with M184V being the most common, present in 12 (1%) individuals, followed by K65R, which was present in 8 (1%) individuals. The combination of at least one NRTI mutation and one NNRTI mutation was identified in 18 (2%) participants.

**Virologic suppression rates.** After excluding 48 individuals who were censored before 12 weeks, virologic success over 96 weeks of observation, as defined by our primary outcome, was achieved in approximately 83% of study participants (678/825, Table 2). In the overall cohort, rates of virologic suppression were significantly lower in those with PDR 65% (73/112) compared to those without PDR (85% [605/713], P < 0.001). This pattern was true for participants initiating EFV-based ART (60% [12/20] vs 86% [214/248], P = 0.002) and DTG-based ART (61/92 [66%] vs 84% [391/465], P value < 0.001, Fig. 3).

In multivariable regression models, PDR remained a strong predictor of virologic success (AOR 0.38, 95%CI 0.21, 0.61) after adjustment for demographic and clinical factors, and both self-reported and pill count-based adherence, as well as additive effects of PDR and adherence in both the EFV and DTG arms (Table 3 and Fig. 4). The effect size and confidence interval estimated would mean that an unmeasured confounder would require an odds ratio of 2.7 or greater with both PDR and virologic suppression (conditional on other confounders, including self-reported adherence) to reduce the effect seen between PDR and virologic success to the null[22]. Viral suppression was also lower in those with higher baseline viral loads and in those with lower self-reported adherence. The effect of PDR did not differ by treatment arm (P value for interaction term by regimen = 0.42). Rates of virologic success were higher for both regimens for those with and without PDR in our secondary outcome, which assessed for persistent virologic failure with two consecutive viral loads >200 copies/mL, although the effect of PDR persisted (85% [73/86] vs 94% 428/453], P = 0.001 for DTG-based ART; 68% [13/19] vs 93% [217/233], P < 0.001 for EFV-based ART, Table 2). The effect of PDR on treatment outcomes persisted as well as for both the FDA 48-week and 96-week Snapshot analyses, including in multivariable analyses (Supplementary information). Among those with the TDF-associated resistance mutation K65R at baseline (n = 8), two were in the EFV arm (both failures) and of the six in the DTG arm, 2/6 (33%) achieved 96-week virologic suppression as defined by the primary outcome measure. Participants with K65R all had NNRTI mutations and 6/8 (75%) had M184V.

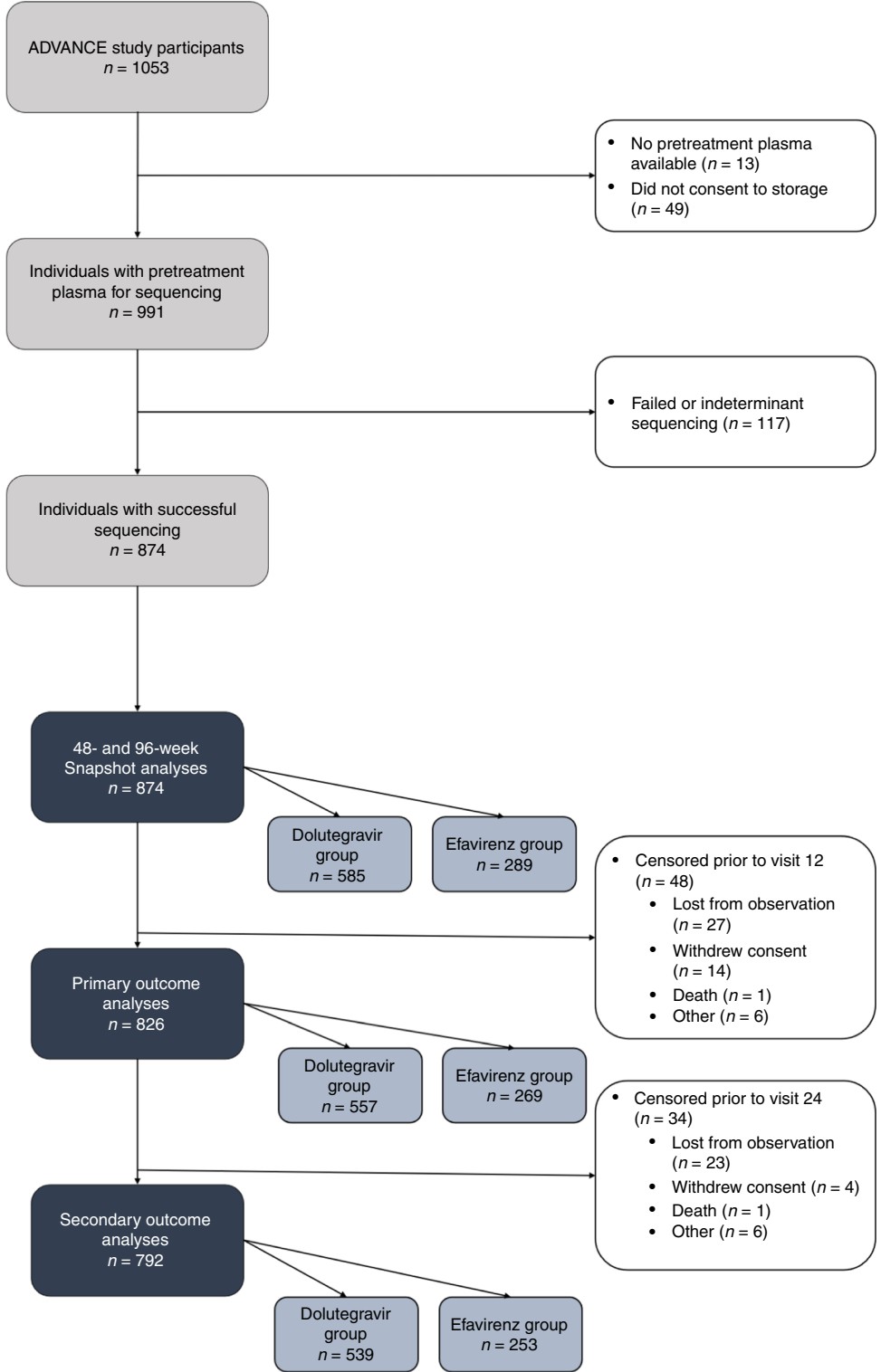

**Fig. 1 Study schema.** Study schema showing numbers of participants with plasma samples, successful sequencing and numbers in primary and secondary outcomes.

In contrast to the effect seen with long-term virologic outcomes, PDR only had an effect on initial virologic response for individuals on EFV-based ART, but not those on DTG-based ART. The change in $\log_{10}$ viral load from enrollment to week 12 was greater for those without PDR in the EFV arm (1.89 vs 2.61 $\log_{10}$ copies/mL, $P < 0.001$), but not in the DTG arms (2.76 vs 2.68 $\log_{10}$ copies/mL, $P < 0.43$, $P = 0.001$ for interaction between arms, Table 2, Supplementary information). In survival analyses, individuals in the EFV arm with PDR experienced a longer time to suppression than whose without PDR ($P = 0.04$ by log-rank testing), whereas those with and without PDR had similar time to initial suppression in the DTG arms ($P = 0.54$ by log-rank testing, Supplementary information). In adjusted Cox proportional hazards models, the effect of PDR remained significant only for

**Table 1 Cohort characteristics for participants who completed pretreatment HIV drug resistance testing and included in our primary analysis of virologic failure, divided by regimen.**

| | Efavirenz arm (n = 269) | Dolutegravir arms (n = 557) | P value[a] |
|---|---|---|---|
| Female sex (n, %) | 153 (56.9%) | 341 (61.2%) | 0.23 |
| Age (median, IQR) | 32 (27−37) | 32 (27−38) | 0.83 |
| Married or partner (n, %) | 60 (22.3%) | 108 (19.4%) | 0.34 |
| Tertiary education (n, %) | 18 (6.7%) | 51 (9.2%) | 0.22 |
| Employed (n, %) | 170 (63.7%) | 349 (63.8%) | 0.97 |
| Pretreatment CD4 count (n, %) | | | 0.58 |
| ≤200 cells/μL | 80 (29.7%) | 179 (32.1%) | |
| 201−350 cells/μL | 81 (30.1%) | 166 (29.8%) | |
| 351−500 cells/μL | 58 (21.6%) | 99 (17.8%) | |
| >500 cells/μL | 50 (18.6%) | 118 (20.3%) | |
| Pretreatment viral load (n, %) | | | 0.33 |
| <10,000 copies/mL | 89 (33.1%) | 171 (30.7%) | |
| 10,000−100,000 copies/mL | 113 (42.0%) | 264 (47.4%) | |
| >100,000 copies/mL | 67 (24.9%) | 122 (21.9%) | |
| Low self-reported adherence[b] (n, %) | 113 (42.0%) | 252 (45.2%) | 0.38 |
| Pill count adherence (n, %)[c] | | | 0.45 |
| <90% | 12 (4.5%) | 33 (6.3%) | |
| 90−95% | 23 (8.6%) | 58 (10.5%) | |
| 95−100% | 233 (87.0%) | 463 (83.6%) | |
| Presence of any WHO-defined pretreatment drug resistance | 20 (7.4%) | 92 (16.5%) | <0.001 |

[a]P values represent statistical tests comparing those included and excluded from the analytic dataset, using chi-squared testing to compare categorical variables and Mann−Whitney nonparametric tests to compare median age.
[b]Low adherence defined as self-report of less than perfect adherence in the 4 days prior to any study visits during the observation period.
[c]Pill count was calculated at each visit by study pharmacists, capped at 100%, then averaged across the 96-week observation period.

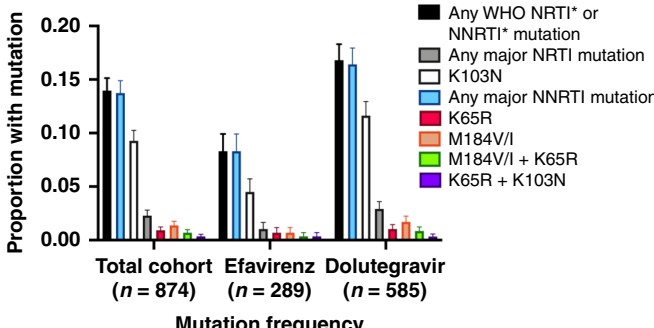

**Fig. 2 Distribution of WHO-defined pretreatment drug resistance in the ADVANCE trial, using the WHO Surveillance Drug Mutations list for mutations detected at >20% of the viral quasispecies.** Error bars indicate 95% confidence intervals around the proportion estimates.

those on EFV-based ART (AHR 0.58, 95%CI 0.35, 0.96), but not for those DTG-based ART (AHR 1.01, 95%CI 0.80, 1.27).

We considered the impact of isolated K103N (as majority virus population, >20%) on virologic response to EFV and DTG (Supplementary information). Rates of virologic suppression were similar in participants taking EFV-based ART with and without the K103N mutation, although the number of individuals with K103N was small in this arm (n = 8). Isolated K103N was associated with lower virologic success for individuals on DTG-based ART, with the exception of our secondary outcome, for which the effect size was similar but the effect was not statistically significant.

We next examined the impact of minority variant PDR in 2−20% of viral quasispecies on outcome of first-line ART. Individuals with mutations in minority populations had similar virologic outcomes as those without PDR overall, and for both those taking DTG- or EFV-based ART (Supplementary

information). We found similar effects of PDR on virologic success in analyses stratified by EFV vs DTG-based treatment (Supplementary information). In a subset of 38 individuals who had sequencing data available both prior to enrollment and at the time of failure, we found that new NRTI and NNRTI mutations developed among those in the EFV-based ART arm in 31% (5/16) and 40% (4/10) of individuals respectively, but that new resistance in the reverse transcriptase gene was rare among those in the DTG-based ART arms (6% [1/17] new NRTI mutations and 0% (0/11) new NNRTI mutations, Supplementary information). Finally, we found no difference in the effect of PDR on any outcomes in sub-analyses restricted to sequences with at least 1000× average depth coverage (Supplementary information).

## Discussion

We report a strong association between drug resistance before treatment initiation, primarily to the NNRTI class, and virologic failure for people initiating first-line ART in the ADVANCE clinical trial. The effect was seen among individuals in the EFV arm and DTG arms, and persisted after adjusting for self-reported and pill count-based adherence and baseline viral load. When we considered a secondary outcome, which focused on persistent virologic failure (two or more consecutive visits with a high viral load), the effect of PDR on DTG persisted, but to a lower degree. In contrast to the effects seen for long-term outcomes, we did not find that PDR had an impact on time to initial suppression or change in quantified viral load from enrollment to 12 weeks, suggesting that NNRTI PDR affects longer-term maintenance of suppression for DTG-based ART or via a non-virally mediated behavioral mechanism. Nonetheless, the finding that NNRTI resistance appears to ultimately predict treatment failure among individuals initiating DTG-based ART in LMIC was unexpected, and to our knowledge not previously reported in the literature.

A virologic mechanism to explain our findings has not been established. NNRTI mutations are not known to affect susceptibility of DTG. The observed effect we identified may be

**Table 2 Virologic success in the ADVANCE Trial by the presence of WHO-defined pretreatment drug resistance.**

| | Total cohort | | | Efavirenz arm | | | Dolutegravir arms | | | Interaction P value[a] |
|---|---|---|---|---|---|---|---|---|---|---|
| | PDR | No PDR | P value[a] | PDR | No PDR | P value[a] | PDR | No PDR | P value | |
| Primary outcome[b] | 73/112 (65%) | 606/714 (85%) | <0.001 | 12/20 (60%) | 215/249 (86%) | 0.002 | 61/92 (66%) | 391/465 (84%) | <0.001 | 0.39 |
| Secondary outcome[c] | 86/105 (82%) | 646/687 (94%) | <0.001 | 13/19 (68%) | 218/234 (93%) | <0.001 | 73/86 (85%) | 428/453 (94%) | 0.001 | 0.26 |
| 48-week Snapshot[d] | 84/122 (69%) | 630/752 (84%) | <0.001 | 11/24 (46%) | 213/265 (80%) | <0.001 | 73/98 (75%) | 417/487 (86%) | 0.006 | 0.10 |
| 96-week Snapshot[d] | 71/122 (58%) | 593/752 (79%) | <0.001 | 11/24 (46%) | 198/265 (75%) | 0.002 | 60/98 (61%) | 395/487 (81%) | <0.001 | 0.64 |
| Mean change in $\log_{10}$ viral load from baseline to 12 weeks (SD) | 2.63 (0.95) | 2.66 (0.79) | 0.78 | 2.00 (0.73) | 2.61 (0.75) | 0.004 | 2.80 (0.94) | 2.68 (0.81) | 0.34 | 0.001 |

[a]P values for the total cohort and treatment arms represent results of chi-squared tests for the primary and secondary outcomes and results of two-sided t tests for the change in viral load outcome. P values for the interaction terms represent two-sided tests of significance for interaction terms between pretreatment drug resistance and efavirenz vs dolutegravir terms in logistic regression models.
[b]Primary outcome: Virologic success in our primary outcome was defined as achievement of a sustained viral load < 1000 copies/mL from 12 weeks, <200 copies/mL from 24 weeks, and <50 copies/mL from 48 weeks onwards. Individuals who are censored after 48 weeks with virologic suppression are considered as achieving virologic success.
[c]Secondary outcome: Virologic success in our secondary outcome was defined as the absence of two consecutive visits with a viral load > 200 copies/mL. Individuals who are censored with a single viral load > 200 copies/mL are considered failures, whereas those who discontinue with virologic suppression are considered as achieving virologic success.
[d]48 and 96-week Snapshot outcome refer to Food and Drug Administration-defined Snapshot outcomes for HIV therapeutic trials.
PDR: presence of WHO-defined pretreatment drug resistance.

consistent with preliminary data suggesting higher replication of NNRTI mutant viruses in the context of drug pressure from integrase inhibitors[23], although additional studies would be needed to corroborate this hypothesis. Whereas we found relatively little minority resistance and no effect of minority resistance on outcomes, existence of NNRTI resistance could be a surrogate marker of archived NRTI resistance[24]. Integrase resistance mutations were not assessed in this study, but are generally believed to be rare (<1%) in this region[25,26]. Importantly, PDR did not hamper time to initial virologic suppression or change in viral load from enrollment to week 12 among people on DTG-based ART. A similar phenomenon of early maintenance of suppression followed by longer-term treatment failure was also seen with DTG monotherapy studies[27,28].

Alternatively, the lack of long-term suppression we identified in the DTG-based ART arms may be due to a behavioral component—pre-existing EFV mutations may be a surrogate of prior default among participants not disclosing previous ART exposure. Our multivariable logistic regression models included a measure of self-reported adherence and pill count-based adherence, both of which were highly predictive of virologic outcomes. Addition of these measures to our model did not meaningfully alter the effect size of PDR on virologic success. However, both self-reported and pill count-based adherence are imperfect measures, and can have a relatively low sensitivity to detect poor adherence, so residual confounding might be present[29–31]. Moreover, previous studies have demonstrated that prior ART exposure is associated with treatment failure and predicts virologic failure, even after controlling for PDR and treatment adherence, and thus could be a source of residual confounding if significant numbers of individuals in this study declined to disclose prior ART use[32,33]. Notably, a number of studies have reported denial of ART use among individuals determined to be taking therapy based on drug level testing[34–36]. The South African public HIV program has used EFV in first-line therapy since its inception in 2004 and has over 5 million on treatment. The number of individuals who have defaulted and are re-initiating therapy are likely to be significant, and it is impossible to identify such individuals within clinical trials using existing South African data systems. Thus, while our effect size for PDR is large, and adjusted for confounding, the possibility that prior treatment exposure is a confounder of this effect remains substantial.

Whether the mechanism of effect is due to poor adherence or virologic mechanisms, our finding that NNRTI resistance, which is present in 10−20% of individuals initiating DTG in the region and is associated with a reduction in efficacy of DTG-based ART, has multiple public health implications. First, ensuring adequate virologic monitoring occurs with DTG-based regimens will remain a priority. Second, treatment programs will require ongoing attention to second- and third-line options, particularly if DTG failure or intolerance becomes more common than previously expected, and NNRTI-based regimens become more commonly used again. Third, integrase resistance testing, which is rarely done outside of research studies in resource-limited settings should become a consideration for referral laboratories in countries where DTG becomes the treatment of choice. Fourth, our data, in combination with the DAWNING study and others, highlight that the presence of drug resistance mutations might portend very different outcomes depending on the timing of when it occurs. In DAWNING, drug resistance mutations detected at the time of first-line failure were a surrogate measure of past adherence and predict success to second-line therapy[21]. By contrast, in this study the presence of drug resistance mutations at presentation for first-line treatment (or re-initiation after a default) appeared to signal the opposite—increased risk of virologic failure, perhaps mediated by poor adherence, or a virologic

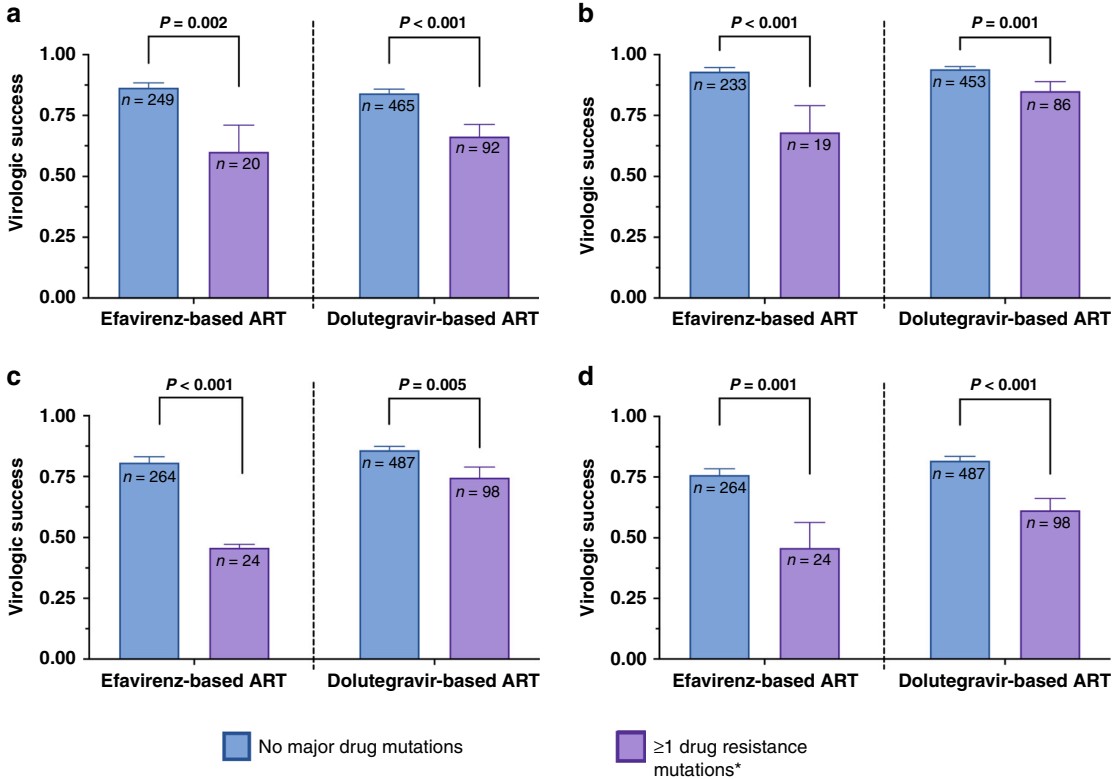

**Fig. 3 Virologic success in the ADVANCE Trial divided by the presence or absence of WHO-defined pretreatment major drug mutations and by use of efavirenz- or dolutegravir-based regimen.** Results are for virologic success defined by our primary outcome (**a**), secondary outcome (**b**), FDA 48-week Snapshot (**c**), and FDA 96-week Snapshot (**d**). Error bars indicate 95% confidence intervals around the proportion estimates. *P* values represent results of two-sided two-proportion *Z* tests.

fitness deficit. Making this distinction about the clinical implications of drug resistance for clinical and public health purposes could be crucial. Fifth, our findings might signal a warning for national programs in the midst of large-scale switching from EFV-based to DTG-based ART, and support increased vigilance for the presence of treatment failure at the time of switch. Finally, these results signal the importance of future work to determine optimal treatment recommendations for individuals failing DTG-based ART, for which minimal data are currently available. Without such data, treatment programs should be advised to maintain virologic monitoring programs, adherence monitoring and support programs for those with failure, and regimen change guidance for individuals with intolerance or persistent virologic failure, even when documented drug resistance is absent. More novel strategies, such as real-time adherence and resistance monitoring, and long-acting injectable formulations of ART for those with adherence challenges should also be explored as these options become more widely available[37–39].

NAMSAL is the only other randomized controlled trial that has compared DTG vs EFV-based first-line ART in sub-Saharan Africa. That study, conducted in Cameroon, compared low-dose 400 mg EFV to DTG as third agent at 48 weeks[6]. DTG was non-inferior to EFV in that study, but baseline VL > 100,00 copies/mL predicted failure in both arms. NAMSAL reported a much lower prevalence of NNRTI resistance (6%) than we did (14%), which is consistent with other data in the region[40]. In NAMSAL, investigators reported no impact of baseline NNRTI resistance on outcomes, although 6/16 failures in the EFV arm had pre-existing NNRTI resistance. In that study, none of the three failures in the DTG arm at 48 weeks had baseline resistance to NNRTIs, the 6% of those that did appear to suppress during the study. By contrast, in our study, isolated K103N in the DTG arm was associated with

lower virologic success in the primary analysis, albeit at 96 weeks. As in prior studies, we identified a very small number of individuals with resistance to both the NRTI and NNRTI drug classes, including K65R, M184V who we believe were unlikely to be treatment naive and who responded poorly to first-line ART. While the proportion is low, this finding is concerning from the point of view of the large-scale EFV to DTG-transition in sub-Saharan Africa, during which multi-class drug resistance is likely to be more prevalent[32,41–44].

Next-generation sequencing is becoming more widely used in research studies to measure the prevalence and impact of drug resistance in LMIC, and has the added advantage of being able to detect resistant viruses at low frequencies[45–47]. However, many studies, and particularly those considering newer ART regimens, have failed to demonstrate a role for these low-level mutant viruses in determining clinical outcomes[48]. We also found no association between PDR and outcome when considering individuals with mutations in between 2 and 20% of viral quasispecies, which supports current practice to use major resistance mutation frequencies for determination of clinically significant drug resistance.

Our study should be generalized in light of its conduct in South Africa, and the presence of NNRTI resistance-conferring mutations as the large majority of the PDR detected. As this is the first study to show an impact of PDR on the efficacy of first-line DTG, it requires corroboration from future studies of similar cohorts. The presence of a higher prevalence of PDR in the DTG arm suggests that there might have been imbalance between groups, which is most likely due to chance, because the study arm was determined by computer randomization. Nonetheless, we have low suspicion for selective dropout in the study because interest in DTG among patients and within society at the time of

**Table 3 Logistic regression models for 96-week virologic success in the ADVANCE Trial (virologic success in our primary outcome was defined as achievement of a sustained viral load <1000 copies/mL from 12 weeks, <200 copies/mL from 24 weeks, and <50 copies/mL from 48 weeks onwards. Individuals who are censored after 48 weeks with virologic suppression are considered as achieving virologic success).**

| Covariable | Univariable models | | Baseline viral load-adjusted multivariable model | | Fully adjusted multivariable model | |
|---|---|---|---|---|---|---|
| | Odds ratio (95%CI) | P value[a] | Adjusted odds ratio (95%CI) | P value[a] | Adjusted odds ratio (95%CI) | P value[a] |
| Female Sex | 0.90 (0.62, 1.29) | 0.59 | | | 0.82 (0.54, 1.25) | 0.35 |
| Age (each year) | 1.05 (1.02, 1.07) | <0.001 | | | 1.02 (0.99, 1.05) | 0.14 |
| Married or partner | 1.38 (0.86, 2.23) | 0.18 | | | 0.95 (0.56, 1.60) | 0.84 |
| Tertiary education | 0.83 (0.45, 1.54) | 0.66 | | | 0.81 (0.41, 1.57) | 0.53 |
| Employed | 2.07 (1.43, 2.98) | <0.001 | | | 1.77 (1.17, 2.67) | 0.01 |
| Pretreatment CD4 count | | | | | | |
| ≤200 cells/μL | REF | | | | REF | |
| 201−350 cells/μL | 1.31 (0.83, 2.07) | 0.25 | | | 1.27 (0.76, 2.13) | 0.37 |
| 351−500 cells/μL | 1.12 (0.67, 1.87) | 0.66 | | | 0.98 (0.54, 1.77) | 0.95 |
| >500 cells/μL | 1.13 (0.68, 1.88) | 0.63 | | | 0.99 (0.54, 1.83) | 0.97 |
| Pre-treatment viral load | | | | | | |
| <10,000 copies/mL | REF | | REF | | REF | |
| 10,000−100,000 copies/mL | 0.59 (0.37, 0.92) | 0.02 | 0.56 (0.36, 0.89) | 0.01 | 0.52 (0.31, 0.88) | 0.01 |
| >100,000 copies/mL | 0.49 (0.29, 0.82) | 0.006 | 0.51 (0.30, 0.85) | 0.01 | 0.39 (0.21, 0.72) | 0.003 |
| Low self-reported adherence[b] (n, %) | 0.36 (0.25, 0.52) | <0.001 | | | 0.41 (0.27, 0.63) | <0.001 |
| Pill count adherence (n, %)[c] | | | | | | |
| <90% | REF | | | | REF | |
| 90−95% | 2.99 (1.39, 6.43) | 0.005 | | | 2.71 (1.15, 6.38) | 0.02 |
| 95−100% | 6.16 (3.31, 11.46) | <0.001 | | | 3.51 (1.70, 7.24) | 0.001 |
| Regimen | | | | | | |
| Efavirenz-based regimen | REF | | REF | | REF | |
| Dolutegravir-based regimen | 0.80 (0.54, 1.18) | 0.26 | 0.90 (0.22, 0.53) | | 1.02 (0.67, 1.57) | 0.92 |
| Presence of WHO-defined pretreatment drug resistance | 0.33 (0.22, 0.52) | <0.001 | 0.34 (0.22, 0.53) | <0.001 | 0.38 (0.23, 0.61) | <0.001 |

[a]P values represent results of two-sided tests of significance for coefficients in multivariable logistic regression models.
[b]Low adherence defined as self-report of less than perfect adherence in the 4 days prior to any study visits during the observation period.
[c]Pill count was calculated at each visit by study pharmacists, capped at 100%, then averaged across the 96-week observation period.

randomization was minimal. Our estimates could be susceptible to unmeasured or residual confounding, particularly due to the effects of prior ART exposure and/or imperfect adherence not captured by self-report or pharmacy pill counts. Notably, our estimates of the effect of PDR on virologic outcomes remained large and significant after adjustment for confounders, including adherence, meaning an unmeasured confounder would have to have a strong association (OR of 2.7 or greater) with both PDR and virologic success to reduce the effect of pretreatment drug resistance to null[22]. Yet, prior studies have shown such an effect size for prior ART exposure[33,49]. Notably, known predictors of treatment success, such as adherence and pretreatment viral load, each predicted virologic success, which enhances the internal validity of our estimates. We also were unable to sequence approximately 15% of the study cohort due to unavailable specimens or failed sequencing. Despite that, our sample size remained large enough to detect relatively small changes in outcomes, and we detected no differences in characteristics between those who were and were not included in this sub-study, which reduces the risk of selection bias. Finally, our sequencing did not include the integrase region of the *pol* gene. Although resistance mutations that confer resistance to dolutegravir remain rare in South Africa, we cannot exclude the possibility of low-level resistance as a possible contributor to our findings[25,26].

In summary, our study suggests that the presence of PDR to NNRTIs is negatively associated with long-term virologic outcome of both EFV- and DTG-based first-line ART in South Africa. In the context of highly prevalent PDR NNRTI resistance, our findings, if corroborated, have implications for first-line ART

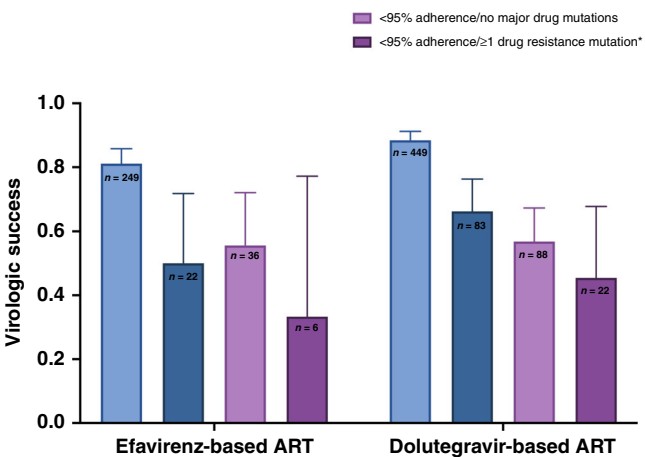

**Fig. 4 96-week treatment outcomes among participants in the ADVANCE Trial divided by treatment arm, presence or absence of WHO-defined pretreatment drug resistance, and achievement of greater than vs less than 95% adherence based on pharmacy pill count.** Error bars represent 95% confidence intervals of the proportion estimates.

selection and treatment monitoring guidelines in the region. Future work should validate our findings, assess the contribution of pretreatment integrase mutations to outcomes, elucidate the impact of prior exposure to ART on treatment outcomes, and

whether treatment failure observed on DTG-based ART is associated with emergence of integrase inhibitor mutations.

## Methods

**Study design**. The ADVANCE trial is an open-label, non-inferiority, phase three clinical trial comparing three regimens for the initial treatment of HIV (NCT03122262). Individuals were recruited from 11 public HIV clinics in Johannesburg, South Africa. All study visits and data collection procedures were performed at one of two research clinics in Johannesburg operated by the study staff. Consenting participants were randomized in a 1:1:1 ratio to (i) tenofovir disoproxil fumarate (TDF), emtricitabine (FTC), EFV; (ii) TDF, FTC, DTG, or (iii) tenofovir alafenamide (TAF), FTC, and DTG[50]. The study enrolled non-pregnant individuals over 12 years old without chronic kidney disease. Individuals were excluded if they had more than 30 days of prior ART use, any ART use in the past 6 months, were pregnant, or were actively undergoing therapy for tuberculosis.

**Study visits and measures**. Study participants were seen for screening and randomization visits, which included collection of blood for pretreatment viral load and CD4 T-cell count measurements. Data on demographics, employment, marital status, and education attainment were collected. During observation, participants were scheduled for visits at week 4, 12, then every 12 weeks thereafter. Data for this analysis are limited to 96 weeks of observation. At each follow-up visit, plasma was collected for viral load estimation. Participants were asked about self-reported adherence over the past 4 days prior to each visit. Finally, study pharmacists recorded dispensed pills and performed a pill count of remaining pills at each follow-up visit. Pretreatment plasma specimens were shipped to KwaZulu-Natal Research Innovation and Sequencing Platform (KRISP) for extraction (Chemagic 360; Perkin Elmer, Germany), HIV-1 pol gene amplification (ThermoFisher HIV-1 genotype amplification module; Life Technologies, CA, USA), and next-generation sequencing (Illumina MiSeq; llumina, CA, USA) as previously described[51]. We limited our analyses to sequences with ≥100× depth of coverage and spanning PR codons 1–99 and RT codons 1–254 (Supplemental Fig. 3).

**Statistical analysis**. We first described and graphically depicted the analytic sample to determine which study participants were included and excluded from this analysis. To assess for selection bias in this sub-analysis, we compared characteristics between individuals who had sequencing results available for this analysis with those who did not due to lack of available plasma specimens or failed sequencing. We then summarized clinical and demographic features of the analytic sample in total, and divided into those initiating EFV and DTG-based regimens. We then described the frequency and proportion of WHO-assigned PDR mutations overall, by drug class, and by treatment regimen.

Our primary exposure of interest was PDR, which we defined as the presence of at least one of the WHO list of surveillance drug mutations detected in at least 20% of the viral population[52]. Our primary outcome of interest was 96-week virologic success, which we defined as sustained a viral load <1000 copies/mL from 12 weeks through 96 weeks, <200 copies/mL from 24 weeks through 96 weeks, and <50 copies/mL from 48 weeks through 96 weeks. Individuals censored with virologic suppression at 48 weeks or after are considered to have achieved virologic success. Individuals who did not complete 12 weeks of observation are not included in this analysis (but are included in the 48- and 96-week Food and Drug Administration [FDA] Snapshot sensitivity outcomes as failures, as described below). We derived this definition to reflect treatment response in individuals who attain and maintain virologic suppression over the course of study observation. We estimated the proportion of participants who achieved virologic suppression by the presence or absence of PDR for the total cohort, and by EFV vs DTG treatment regimens.

We fitted logistic regression models with virologic success as the outcome of interest to estimate the contributions of both PDR and regimen to 96-week virologic success, with and without a regimen by PDR product interaction term to assess whether the effect of PDR differed by EFV vs DTG use. We then fitted multivariable logistic regression models with virologic success as the outcome of interest, including the following potential confounding variables, which have been shown to determine virologic success in prior work[53,54]: sex, age, partnership status (defined as married or with a primary partner vs not), educational attainment (dichotomized as tertiary education or less), active employment status, pretreatment CD4 T-cell count (categorized as ≤200 cells/μL, 201−350 cells/μL, 351−500 cells/μL, and >500 cells/μL), pretreatment viral load (categorized as <10,000 copies/mL, 10,000−100,000 copies/mL, >100,000 copies/mL), pill count-based adherence (calculated as the number of pills taken since the prior visit divided by the expected number of pills taken, capped at 100% at each visit, averaged over the course of the 96-week observation period, and categorized as 95−100%, 90−95%, and <90% average adherence), and self-reported adherence (dichotomized as perfect adherence in the past 4 days vs any treatment interruptions in the past 4 days). We assessed for collinearity in our model by estimating variation initiation factors for each covariate within our fully adjusted model (Supplemental Table 7). In a third model intended to focus more directly on the virologic factors that determine treatment outcome, we restricted the model to pretreatment viral load, presence or absence of PDR, and study treatment allocation.

In sensitivity analyses, we varied our definition of virologic success. To assess for the impact of NNRTI PDR on persistent virologic failure, for a secondary

outcome we defined success in individuals without two consecutive visits up to 96 weeks with a viral load >200 copies/mL. In this definition, individuals censored or changed to a second-line regimen after a single viral load >200 copies/mL are considered failures, whereas those who discontinue with virologic suppression are considered as achieving virologic success. This outcome is meant to detect persist virologic failure by allowing for virologic blips or episodic failure followed by re-suppression. To assess the contribution of PDR on virologic response to therapy, we also considered a virologic potency outcome defined by the change in $\log_{10}$ viral load from enrollment to week 12, and assessed for time to first virologic suppression using Kaplan−Meier survival methods. In the survival analyses, individuals were censored at the time of first virologic suppression and considered failures if that occurred with a detectable viral load. We also conducted analyses using the FDA-defined 48-week and 96-week Snapshot to define virologic success. In these analyses, individuals who dropped out prior to the 48- and 96-week windows are considered as failures, irrespective of the reason. We considered three stratified analyses in which we (1) restricted the definition of PDR to individuals with only the K103N mutation, (2) restricted the definition of PDR to individuals who had WHO-defined PDR mutations at variant frequencies of 2−20%, and (3) assessed finding stratified by those in the EFV- or DTG-based arms. In a subset of individuals who had sequencing data available both prior to treatment initiation and after failure, we compared changes in patterns of NRTI and NNRTI-class resistance over time. We also conducted restricted analysis to individuals with sequences with ≥100× depth of coverage. Finally, we estimate an E value to determine the magnitude of the effect size of an unmeasured confounder who need to have to reduce the association between PDR and virologic success to null[22]. Data analysis was conducted in Stata (Version 15.1, Statacorp, College Station, Texas, USA), coded by two separate investigators (M.J.S. and B.S.) and compared for reproducibility. A fully de-identified dataset and code for all analyses are available upon request to the corresponding author.

**Ethical considerations**. The study was approved by the institutional review board at the University of the Witwatersrand. All study participants gave written informed consent to participate.

**Reporting summary**. Further information on research design is available in the Nature Research Reporting Summary linked to this article.

## Data availability

Full data are available from Professor Ravi Gupta (rkg20@cam.ac.uk) or Professor Francois Venter (fventer@ezintsha.org). Sequences generated in this study are available from SRA: https://www.ncbi.nlm.nih.gov/sra under accession number: PRJNA669549.

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

## Acknowledgements

We thank the outstanding staff at Ezintsha (a sub-division of the Wits Reproductive Health and HIV Institute) and the Africa Health Research Institute for their work in conducting the study, as well as the community and the study participants for their time and efforts. We thank Tijana Stanic for her contributions to figure design and construction. The ADVANCE trial was supported by USAID, Unitaid, the South African Medical Research Council (SAMRC), with investigational drug donated by ViiV Healthcare and Gilead Sciences. R.K.G. is supported by a Wellcome Trust Fellowship (WT108082AIA). M.J.S. is supported by the National Institutes of Health (R01 AI124718, R01 AG059504)

## Author contributions

Conceptualization: ideas; formulation or evolution of overarching research goals and aims: M.J.S., R.K.G., W.D.F.V., A.H. Data curation: Management activities to annotate (produce metadata), scrub data and maintain research data (including software code, where it is necessary for interpreting the data itself) for initial use and later reuse: M.J.S., B.S., G.A., J.G., B.C., M.A.M., R.L. Formal analysis: Application of statistical, mathematical, computational, or other formal techniques to analyze or synthesize study data: M.J.S., B.S. Funding acquisition: Acquisition of the financial support for the project leading to this publication: A.H., W.D.F.V. Investigation: Conducting a research and investigation process, specifically performing the experiments, or data/evidence collection: M.A.M., G.A., C.M.S., W.D.F.V., A.H. Methodology: Development or design of methodology; creation of models: A.H.,

W.D.F.V., M.J.S., R.K.G., T.d.O., R.L. Validation: Verification, whether as a part of the activity or separate, of the overall replication/reproducibility of results/experiments and other research outputs: M.J.S., B.S., T.d.O., R.L., B.C., J.G., G.A., S.A.K. Visualization: Preparation, creation and/or presentation of the published work, specifically visualization/data presentation: M.J.S. Writing—original draft preparation: Creation and/or presentation of the published work, specifically writing the initial draft (including substantive translation): M.J.S. Writing—review and editing: Preparation, creation and/or presentation of the published work by those from the original research group, specifically critical review, commentary or revision—including pre- or post-publication stages: M.J.S., M.A.M., B.S., T.d.O., R.L., J.G., S.A.K., B.C., G.A., C.M.S., W.D.F.V., A.H., R.K.G.

## Competing interests

R.K.G. has received ad hoc consulting fees from Gilead, ViiV and UMOVIS Lab. W.D.F.V. received drug donations from ViiV Healthcare and Gilead Sciences for investigator-led clinical studies, including ADVANCE. In addition, he receives honoraria for talks and board membership for: Gilead, ViiV, Mylan, Merck, Adcock-Ingram, Aspen, Abbott, Roche, J&J. M.A.M. received drug donations from ViiV Healthcare and Gilead Sciences for investigator-led clinical studies, including ADVANCE. In addition, she received honoraria for talks and board membership for: Gilead, ViiV, Mylan, Aspen, AbbVie, Johnson & Johnson, Sanofi, Pfizer and Southern African HIV Clinicians Society. She also received meeting/conference sponsorship from Johnson and Johnson, BD, Gilead, Merck, Cipla, Mylan and Canopy Growth. No other authors have any conflict of interest.
