## [Peer Review File · Nature Communications]

REVIEWER COMMENTS

Reviewer #1 (Remarks to the Author):

Thank you for the opportunity to review this manuscript highlighting novel findings on the impact of NNRTI drug resistance mutations on integrase-based inhibitors. Although the exact mechanism is not elucidated, these findings are critical as they indicate that DTG-based ART may not fully address the challenge of rising NNRTI resistance especially in low- and middle-income countries. In as such there is a dire need to adopt the suggestions highlighted by the authors as well as determining the exact mechanism of how NNRTI DRMs impact the efficacy of DTG-based ART.

Major comments

1. While efforts have been taken to address any potential confounders, it is still possible as the authors have indicated that factors such as prior ARV use and unmeasured adherence could still have impacted the outcome of the study. While this is also addressed in the discussion, it would still be good to note that the highlighted effect size of an unmeasured confounder (aOR 2.8) is still possible. The study by Hamers et al observed that prior ARV use was an independent predictor of virological failure even after controlling for both PDR and non-adherence with an effect size of 3.10. In a further analysis of the same study it was noted that the effect of prior ARV use that could be addressed by eliminating the effect of PDR was only 48% suggesting a residual effect that was mediated possibly through non-adherence or an unknown mechanism. However, like in this study imperfect measures of adherence were used. Perhaps these findings may suggest the need to use more objective measures of adherence and alternative more objective tools to assess prior ARV use.
2. In addition to above, it is indeed likely that these findings may be due to a behavioral mechanism. Although the authors cite the DAWNING study to support their findings, it is worth noting that the efficacy observed in DAWNING is similar to that observed in the participants without NNRTI DR in the ADVANCE trial. Moreover, a sub-analysis in DAWNING showed that participants with multiple NRTI resistance i.e. M184V+ TAMs had even much better outcomes (noting that they were also likely to have NNRTI resistance). Although this does not rule out the benefit of hypersensitization of the viral mutants to the alternative NRTIs, it may still likely indicate that these patients are likely to be more adherent than those without multiple NRTIs as has been previously postulated. In addition, and as the authors have noted, none of the patients with viral non-suppression in NAMSAL had baseline DR although the overall PDR prevalence was low. Overall it may be good for the authors to discuss the findings in light of both DAWNING and NAMSAL but there is still a need for more studies including in-vitro studies to assess for this phenomenon
3. The authors suggest the need of integrase resistance testing in light of their findings. However previous analysis of the ADVANCE study showed rare occurrence of DTG resistance and no incident

NNRTI resistance. First it would be good to also present or if possible, discuss on incident DR especially if there is any accumulation of either NNRTI, NRTI or INI resistance. Second, the authors could guide on what would be the best strategy to use, if DTG resistance is still rare among patients failing treatment, including those who had NNRTI PDR as it was the case in the ADVANCE trial.

Minor comments

1. Typo in line 276 "in has multiple public health implications" Suggest to remove "in"

Reviewer #2 (Remarks to the Author):

This is an interesting analysis suggesting that the NNRTI associated resistance detected at >20% of the virus population is associated to 96-week virological failure of DTG-based first line regimen in the ADVNCE trial. If true, this could have serious implications for treatment management of HIV-infected people in the resource limited settings and elsewhere. Although the paper is fairly well written, the analysis and the arguments in favour of the set hypothesis are fairly unconvincing. I summarise my main points below.

Main Points

1. Exposure. Lines 131-134. Sequence read coverage depth is likely to vary markedly across the sequenced amplicons (highest in protease than in RT). Which minimum read depth was chosen and spanning across which codons in each region? This might have been described in the parental trial paper but I think that it is crucial to repeat here. Do results vary by using different coverage depths?
2. Virological outcome. The authors used 3 separate virological endpoints. None of these seems suitable for an analysis looking at virological potency and association with drug resistance. As by the primary outcome someone still with a viral load of say 900 copies/mL by 3 months from starting ART is considered as a success if viral load declines <200 copies/mL by 6 months and <50 copies/mL by one year regardless of treatment switch. Most people achieve a VL<50 by 3 months on modern ART so unclear why these early failures are ignored. The secondary outcome is unusual and unclear what is trying to achieve. People with a single viral load >200 copies/mL if followed by a treatment change should be clearly defined as failures, not censored. By snapshot analysis, treatment discontinuations are all considered as failures regardless of the reason for stopping. These include toxicity and simplifications which are not relevant for a question related to virological potency. The more standard

analyses in this setting are i) change in viral load at month 3 modelled as a continuous variable censoring people who stopped/switched drugs because of toxicity or simplification before month 3 or ii) a time to virological failure analysis (either pure virological failure using two consecutive values >200 copies/mL or a single value followed by change in treatment). I'd like to see the results after these are used.

3. Confounding #1. A factor to be a confounder had to be a predictor of the outcome but also somewhat at least weakly causing pre-treated drug resistance. Most of the variables included in the logistic regression model do not satisfy this definition. In fact, the only important confounder is baseline viral load. Although including predictors of outcome should increase efficiency, it might be beneficial to have a more parsimonious model to avoid overfitting as confounding is unlikely to be introduced by the selected factors. Indeed, Supplemental Table 5 shows that some of these factors (i.e. sex age, education etc.) are quite weakly associated with the outcome. In addition, the model is adjusted for current measures of adherence such as pill count-based adherence (calculated as the number of pills taken since the prior visit and self-reported adherence (dichotomized as perfect adherence in the past 4 days). These are clearly mediators, not confounding factors as pre-existing resistant viruses become dominant via poor adherence after baseline. This is particularly true for EFV which is known to have a lower barrier to resistance than DTG so controlling for adherence would have removed an important part of the total effect. Personally I would only control for baseline viral load and NRTI resistance (both Sanger and 2-20% threshold in separate models) and use these models to test for the interaction.

4. Confounding #2. Authors are clearly aware of the possibility of unmeasured confounding and I was impressed by the calculation of the E-value. However, INSTI resistance (both low level and >20% level) is clearly an unmeasured confounder that could explain the high rate of failure in the DTG arm. For example, I have anecdotal evidence that G140ACRS in INSTI region even if detected at 2-20% level can increase the risk of failing INSTI-based therapy by 60%. Furthermore, there is confounding by NRTI resistance that has not been accounted for in the analysis. Figure 2 clearly shows that both M184V and K65R were more prevalent in people starting DTG. M184I should be also counted in as it confers the same reduction in susceptibility of FTC as M184V. Again, M184VI at 2-20% can be shown to be associated with 3-fold increase in risk of virological failure of triple combination including 3TC/FTC plus an anchor drug including INSTI, a value which exceeds the calculated E-value making residual confounding a possible explanation for the findings.

5. Interaction. The analysis was clearly not powered to test this interaction so a p-value<0.05 for the test does not mean that interaction can be ruled out. Looking at some of the analyses shown in Table 2 the risk difference by PDR in EFV are much larger (almost a 3-fold difference) than those in DTG. For example for the secondary outcome (93%-68% = 25%) for EVF vs. (94%-85%=9%) for DTG and similarly for the 48-week snapshot (81%-46%=35%) for EFV vs. (86%-74%=12) for DTG. On the basis of these differences and all the caveats described above I do not think that the data support that the association between PDR and risk of failure does not vary by treatment arm.

6. Lines 242-243. NNRTI resistance detected at 2-20% of virus population has been consistently shown to be associated with failure of EFV-based regimens. [Casadellà M, Paredes R. Deep sequencing for HIV-1 clinical management. *Virus Res.* 2017 Jul 15;239:69-81]. The fact that these results could not be replicated here, shades doubts on the credibility of the rest of the findings.

Other points

1. Lines 197-199. At the time of data extraction, all 873 had completed observation up to 96 weeks. A total of 48 and 91 individuals were excluded from the primary and secondary analyses, respectively, for not remaining in the study to 12 or 24 weeks. These two sentences seem to contradict each other?

2. Line 207. Typo.individual had a nucleoside, not 'an'

3. Lines 221 and 320. Was the E-value equal to 2.9 or 2.8 ? Which exact analysis was used to calculate this (primary endpoint?)

4. Line 248. We report a strong and pervasive association between NNRTI resistance...Although the majority of participants harboured NNRTI resistance the exposure was defined as having ≥ 1 mutations in the WHO list.

5. Lines 256-258. The authors use the paper by Clutter DS et al in support of their hypothesis that NNRTI resistance might impair response to INSTI-based regimens. The argument seems very speculative especially because the paper actually supports the opposite (INSTI-based regimens are an effective option with at least equal efficacy compared with bPIs for patients with isolated NNRTI TDR).

6. Lines 266-268. Current adherence is a mediator, not a confounder. Although there is suspicion that some may not be ART-naïve patients, clearly the temporality goes against believing that these measures could be confounders.

7. Line 276. Typo. DTG based ART has, not 'in has'

8. Line 307. ...between 5-20% of viral quasiespecies. In the Results section it says 2-20%, which one is correct? The choice would impact on the prevalence of exposure and therefore on the results. Indeed, it is possible that statistical power is lower for the 5-20% threshold vs. 2-20% threshold.

9. Lines 314-315. ...because study arm was determined by computer randomisation. I do not understand this. Exchangeability created by randomisation was broken down by the fact that plasma samples were not available in all participants.

10. Table 2. I would remove the p-values within the strata as these are subset analyses so difficult to interpret.

Reviewer #3 (Remarks to the Author):

The authors assess the efficacy of HIV-1 integrase inhibitors in patients with drug resistance mutations in reverse transcriptase using samples from the ADVANCE trial. Some comments regarding the statistical analyses:

1. The definition of virologic success is a little confusing. If someone achieved a viral load (VL) of <1000 but ≥ 200 from 12 weeks onward considered a success for the entire time period? I understand that the authors would like to simplify the outcome so there is only one outcome per person, but if this participant then had a VL ≥ 200 at 24 weeks were they still considered a success?
2. With the longitudinal nature of the data one definition of viral suppression could be used and a time to event analysis done to look at time to viral suppression. Did the authors consider this?
3. With the outcome (viral suppression) being very common, log binomial regression should probably be used instead of logistic regression to give risk ratios instead of odds ratios.
4. The authors assessed if there were differences in clinical or demographic characteristics between those who successfully underwent sequencing and those who did not. The same should be done to compare the 48 and 91 individuals who did not remain in the study for 12 or 24 weeks.
5. Table 1 compares characteristics by regimen for those with successful sequencing. Given the primary analysis does not include the 48 individuals who did not remain in the study for 12 weeks, it seems that these individuals should be excluded from this table.
6. Table 3: There are a lot of variables in the multivariable model and some of these are likely to be collinear (e.g., low self reported adherence and pill count adherence). Please assess collinearity of all variables and adjust the analyses accordingly.

REVIEWER COMMENTS

Reviewer #1 (Remarks to the Author):

Thank you for the opportunity to review this manuscript highlighting novel findings on the impact of NNRTI drug resistance mutations on integrase-based inhibitors. Although the exact mechanism is not elucidated, these findings are critical as they indicate that DTG-based ART may not fully address the challenge of rising NNRTI resistance especially in low- and middle-income countries. In as such there is a dire need to adopt the suggestions highlighted by the authors as well as determining the exact mechanism of how NNRTI DRMs impact the efficacy of DTG-based ART.

Response

We thank the reviewer for their thorough review and agree with the importance of defining the contributions of drug resistance on treatment outcomes in low and middle-income countries.

Major comments

1. While efforts have been taken to address any potential confounders, it is still possible as the authors have indicated that factors such as prior ARV use and unmeasured adherence could still have impacted the outcome of the study. While this is also addressed in the discussion, it would still be good to note that the highlighted effect size of an unmeasured confounder (aOR 2.8) is still possible. The study by Hamers et al observed that prior ARV use was an independent predictor of virological failure even after controlling for both PDR and non-adherence with an effect size of 3.10. In a further analysis of the same study it was noted that the effect of prior ARV use that could be addressed by eliminating the effect of PDR was only 48% suggesting a residual effect that was mediated possibly through non-adherence or an unknown mechanism. However, like in this study imperfect measures of adherence were used. Perhaps these findings may suggest the need to use more objective measures of adherence and alternative more objective tools to assess prior ARV use.

Response

This is an excellent point. Although our models are adjusted for two different measures of adherence (self-report and pharmacy pill count), we agree that these are imperfect measures and the possibility of residual confounding persists. We thank you for directing us to the work detailing the large effect of prior treatment on outcomes and have added these citations and themes to the discussion section:

Our multivariable logistic regression models included a measure of self-reported adherence and pill count-based adherence, both of which were highly predictive of virologic outcomes. Addition of these measures to our model did not meaningfully alter the effect size of PDR on virologic success. However, both self-reported and pill count-based adherence are imperfect measures, and can have a relatively low sensitivity to detect poor adherence, so residual confounding might be present.¹⁻³ Moreover, prior studies have demonstrated that prior ART exposure is associated with treatment failure and predicts virologic failure, even after controlling for PDR and treatment adherence, and thus could be a source of residual confounding if significant numbers of individuals in this study declined to disclose prior ART use.^{4,5} Notably, a number of studies have reported denial of ART use among individuals determined to be taking therapy based on drug level testing.⁶⁻⁸ The South African public HIV program has used EFV in first line therapy since its inception in 2004 and has over 5 million on

treatment. The number of individuals who have defaulted and are re-initiating therapy are likely to be significant, and it is impossible to identify such individuals within clinical trials using existing South African data systems. Thus, while our effect size for PDR is large, and adjusted for confounding, the possibility that prior treatment exposure is a confounder of this effect remains substantial.

And to the limitations section:

Our estimates could be susceptible to unmeasured or residual confounding, particularly due to effects of prior ART exposure and/or imperfect adherence not captured by self-report or pharmacy pill counts. Notably, our estimates of the effect of PDR on virologic outcomes remained large and significant after adjustment for confounders, including adherence, meaning an unmeasured confounder would have to have a strong association (OR of 2.8 or greater) with both PDR and virologic success to reduce the effect of pre-treatment drug resistance to null.⁹ Yet, prior studies have shown such an effect size for prior ART exposure.^{5,10}

2. In addition to above, it is indeed likely that these findings may be due to a behavioral mechanism. Although the authors cite the DAWNING study to support their findings, it is worth noting that the efficacy observed in DAWNING is similar to that observed in the participants without NNRTI DR in the ADVANCE trial. Moreover, a sub-analysis in DAWNING showed that participants with multiple NRTI resistance i.e. M184V+ TAMs had even much better outcomes (noting that they were also likely to have NNRTI resistance). Although this does not rule out the benefit of hypersensitization of the viral mutants to the alternative NRTIs, it may still likely indicate that these patients are likely to be more adherent than those without multiple NRTIs as has been previously postulated. In addition, and as the authors have noted, none of the patients with viral non-suppression in NAMSAL had baseline DR although the overall PDR prevalence was low. Overall it may be good for the authors to discuss the findings in light of both DAWNING and NAMSAL but there is still a need for more studies including in-vitro studies to assess for this phenomenon

Response

We agree with the reviewer that, in combination, these studies suggest that presence of drug resistance mutations are distinct phenomena depending on when they occur. We have added text to the discussion section to better elucidate this important distinction:

Our data, in combination with the DAWNING study and others, highlight that the presence of drug resistance mutations might portend very different outcomes depending on the timing of when it occurs. In DAWNING, drug resistance mutations detected at the time of first-line failure were a surrogate measure of past adherence and predict success to second-line therapy. By contrast, the ADVANCE study appears to suggest that presence of drug resistance mutations at the time of presentation for first-line treatment (or re-initiation after a default) might signal the opposite – increased risk of virologic failure, perhaps mediated by poor adherence, or a virologic fitness deficit. Making this distinction both for clinical and public health purposes could be crucial.

3. The authors suggest the need of integrase resistance testing in light of their findings. However previous analysis of the ADVANCE study showed rare occurrence of DTG resistance and no incident NNRTI resistance. First it would be good to also present or if possible, discuss on incident DR especially if there is any accumulation of either NNRTI, NRTI or INI resistance. Second, the authors could guide on

what would be the best strategy to use, if DTG resistance is still rare among patients failing treatment, including those who had NNRTI PDR as it was the case in the ADVANCE trial.

Response

We appreciate this suggestion and, although our sample size for on-treatment sequencing is relatively small, we have added an exploratory analysis to compare resistance patterns prior to treatment versus at the time of persistent failure. A total of 38 individuals have both pre-treatment and on-treatment sequencing available in the dataset. Of these, 17 were in the EFV arm and 21 were in a DTG arm. We have added this information to the manuscript in the results section and as supplemental Table 6.

Finally, in a subset of 38 individuals who had sequencing data available both prior to enrolment and at the time of failure, we found that new NRTI and NNRTI mutations developed among those in the EFV-based ART arm in 31% (5/16) and 40% (4/10) of individuals respectively, but that new resistance in the reverse transcriptase gene was rare among those in the DTG-based ART arms (6% [1/17] new NRTI mutations and 0% (0/11) new NNRTI mutations, Supplemental Figure 6).

Supplemental Table 6. Comparison of pre- versus on-treatment WHO-defined reverse transcriptase drug resistance mutation patterns in the entire cohort, and by treatment group

NRTI Mutations											
Total Cohort				Efavirenz Arm				Dolutegravir Arms			
	Pre-treatment Sequencing				Pre-treatment Sequencing				Pre-treatment Sequencing		
On Treatment Sequencing	No NRTI Resistance	NRTI Resistance	Total	On Treatment Sequencing	No NRTI Resistance	NRTI Resistance	Total	On Treatment Sequencing	No NRTI Resistance	NRTI Resistance	Total
No NRTI Resistance	27 (82%)	0 (0%)	27	No NRTI Resistance	11 (69%)	0 (0%)	11	No NRTI Resistance	16 (94%)	0 (0%)	16
NRTI Resistance	6 (18%)	5 (100%)	11	NRTI Resistance	5 (31%)	1 (100%)	6	NRTI Resistance	1 (6%)	4 (100%)	5
Total	33	5	38	Total	16	1	17	Total	17	4	21

NNRTI Mutations											
Total Cohort				Efavirenz Arm				Dolutegravir Arms			
	Pre-treatment Sequencing				Pre-treatment Sequencing				Pre-treatment Sequencing		
On Treatment Sequencing	No NNRTI Resistance	NNRTI Resistance	Total	On Treatment Sequencing	No NNRTI Resistance	NNRTI Resistance	Total	On Treatment Sequencing	No NNRTI Resistance	NNRTI Resistance	Total
No NNRTI Resistance	17 (81%)	2 (12%)	19	No NNRTI Resistance	6 (60%)	0 (0%)	6	No NNRTI Resistance	11 (100%)	2 (20%)	13
NNRTI Resistance	4 (19%)	15 (88%)	19	NNRTI Resistance	4 (40%)	7 (100%)	11	NNRTI Resistance	0 (0%)	8 (80%)	8
Total	21	17	38	Total	10	7	17	Total	11	10	21

We also agree with the importance to derive guidelines with these results in consideration, but are mindful of the limited data available on optimal management of virologic failure on DTG-based ART. We highlight the limitations of the current data and, in response to reviewer suggestion, propose a strategy in the absence of such data:

Finally, these results signal the importance of future work to determine optimal treatment recommendations for individuals failing DTG-based ART, for which minimal data is currently available. Without such data, treatment programs should be advised to maintain virologic monitoring programs, adherence monitoring and support programs for those with failure, and be mindful of the importance of regimen change guidance for individuals with intolerance or persistent virologic failure, even when documented drug resistance is absent. More novel strategies, such as real-time adherence and resistance monitoring, ,and long-acting injectable formulations of ART for those with adherence challenges should also be explored as these options become more widely available.¹¹⁻¹³

Minor comments

1. Typo in line 276 “in has multiple public health implications” Suggest to remove "in"

Response

We have removed the word “in” as suggested.

Reviewer #2 (Remarks to the Author):

This is an interesting analysis suggesting that the NNRTI associated resistance detected at >20% of the virus population is associated to 96-week virological failure of DTG-based first line regimen in the ADVNCE trial. If true, this could have serious implications for treatment management of HIV-infected people in the resource limited settings and elsewhere. Although the paper is fairly well written, the analysis and the arguments in favour of the set hypothesis are fairly unconvincing. I summarise my main points below.

Main Points

1. Exposure. Lines 131-134. Sequence read coverage depth is likely to vary markedly across the sequenced amplicons (highest in protease than in RT). Which minimum read depth was chosen and spanning across which codons in each region? This might have been described in the parental trial paper but I think that it is crucial to repeat here. Do results vary by using different coverage depths?

Response

Thank you for this comment. We included only sequences with $\geq 100X$ depth of coverage and spanning PR codons 1 – 99 and RT codons 1 – 254. We did not observe any variation in results with different coverage depths at $\geq 100X$ and there was no preferential amplification of either gene target (i.e. PR and RT). We have updated the manuscript with this information:

We limited our analyses to sequences with $\geq 100X$ depth of coverage and spanning PR codons 1 – 99 and RT codons 1 – 254.

We added distribution of read depth coverage among samples included in final analysis in Supplemental Figure 3.

Supplemental Figure 3. Distribution of read depth coverage among samples included in the analysis

To assess for the possibility that low coverage depth might have affected our results, we conducted an additional sensitivity analysis, excluding sequences with a coverage depth <1000x, and found no difference in the effect of PDR on any of our outcomes (Supplemental Table 8):

Supplemental Table 8. Virologic success in the ADVANCE Trial by the presence of WHO-defined pretreatment drug resistance in the entire analytic dataset and restricted to sequences with an average coverage depth >1,000

	Total Cohort			Restricted to Average Coverage Depth >1,000		
	PDR	No PDR	P-value	PDR	No PDR	P-value
Primary Outcome ^b	73/112 (65%)	606/714 (85%)	<0.001	67/102 (66%)	560/659 (85%)	<0.001
Secondary Outcome ^c	86/105 (82%)	646/687 (94%)	<0.001	79/96 (82%)	596/634 (94%)	<0.001
48-week Snapshot ^d	84/122 (69%)	630/752 (84%)	<0.001	76/112 (68%)	578/692 (84%)	<0.001
96-week Snapshot ^d	71/122 (58%)	593/752 (79%)	<0.001	64/112 (57%)	546/692 (79%)	<0.001
Mean change in log ₁₀ viral load from baseline to 12 weeks (SD)	2.63 (0.95)	2.66 (0.79)	0.78	2.58 (0.12)	2.65 (0.80)	0.50

2. Virological outcome. The authors used 3 separate virological endpoints. None of these seems suitable for an analysis looking at virological potency and association with drug resistance. As by the primary outcome someone still with a viral load of say 900 copies/mL by 3 months from starting ART is considered as a success if viral load declines <200 copies/mL by 6 months and <50 copies/mL by one year regardless of treatment switch. Most people achieve a VL<50 by 3 months on modern ART so unclear why these early failures are ignored. The secondary outcome is unusual and unclear what is trying to achieve. People with a single viral load >200 copies/mL if followed by a treatment change should be clearly defined as failures, not censored. By snapshot analysis, treatment discontinuations are all considered as failures regardless of the reason for stopping. These include toxicity and simplifications which are not relevant for a question related to virological potency. The more standard analyses in this

setting are i) change in viral load at month 3 modelled as a continuous variable censoring people who stopped/switched drugs because of toxicity or simplification before month 3 or ii) a time to virological failure analysis (either pure virological failure using two consecutive values >200 copies/mL or a single value followed by change in treatment). I'd like to see the results after these are used.

Response

We thank the reviewer for these comments and suggestions. We agree that there are multiple considerations in defining treatment failure that have import. We agree that anti-viral potency, as the reviewer suggests, is a critical outcome and one that we did not thoroughly address in our initial submission. We also feel strongly that a public health approach will also include focus on a definition of long-term maintenance of suppression. Our intention is to explore each of these perspectives to enable consideration of these varying outcomes and their implications. In response, we have taken the following steps to address the weaknesses identified by the reviewer:

- a. We have added a new outcome as suggested by the reviewer, defined as the change in log₁₀ viral load from baseline to 3 months (12 weeks), and censoring those who discontinued prior to week 12 (allowing for a maximum of a 4 week window in those who did not present exactly at 12 weeks). Given the continuous nature of this outcome and its normal distribution, we used studentized t-tests and fit linear regression models to assess this outcome, with the following results:

The viral load reduction between baseline and week 12 was reduced for participants the efavirenz arm (-0.72, 95%CI 0.37, 1.08, P<0.001) but not for those in the dolutegravir arm (-0.08, 95%CI -0.27, 0.11, P=0.43) and the effect of PDR on virologic response was significantly different by arm (Table 2, Supplemental Figure 1):

	Total Cohort			Efavirenz arm			Dolutegravir arms			Interaction P-value
	PDR	No PDR	P-value	PDR	No PDR	P-value	PDR	No PDR	P-value	
Mean change in log ₁₀ viral load from baseline to 12 weeks (SD)	2.63 (0.95)	2.66 (0.79)	0.78	2.00 (0.73)	2.61 (0.75)	0.004	2.80 (0.94)	2.68 (0.81)	0.34	0.001

Supplemental Figure 1. Change in log₁₀ viral load form the baseline enrolment visit to week 12 by treatment arm and presence or absence of World Health Organization-defined pre-treatment drug adherence mutations to reverse transcriptase.

b. We have modified our secondary outcome of persistent virologic failure to match the suggested definition of virologic failure as suggested by the reviewer, defined as two consecutive viral loads >200 copies/mL or a change to a second-line regimen among those with a high viral load. Only two participants changed to a second line regimen due to virologic failure (one in each arm) and as such these alterations had no substantial effect on our persistent virologic failure (secondary) outcome.

c. We have added a time to virologic suppression survival analysis.

Similar to the virologic potency analysis, this time to suppression analysis did show an effect of PDR on time to initial suppression for the EFV ($P=0.04$ by log-rank testing) but not for the DTG arms ($P=0.54$ for log-rank testing):

These results suggest that the presence of PDR does not appear to impact early virologic response to DTG, but does appear to predict risk of longer-term treatment failure among those who initially suppress. We have updated our discussion to discuss the implications of these additional findings:

Results:

In contrast to the long-term virologic failure outcomes, PDR only had an effect on initial virologic response for individuals on EFV-based ART, but not those on DTG-based ART. The change in log₁₀ viral load from enrolment to week 12 was greater for those without PDR in the EFV arm (1.89 vs 2.61 log₁₀ copies/mL, $P<0.001$), but not in the DTG arms (2.76 vs 2.68 log₁₀ copies/mL, $P<0.43$, $P=0.001$ for interaction between arms, Table 2, Supplemental Figure 1). Similarly, in survival analyses, individuals in the EFV arm with PDR experienced a longer time to suppression than those without PDR ($P=0.04$ by log-rank testing), whereas those with and without PDR had similar time to initial suppression in the DTG arms ($P=0.54$ by log-rank testing, Supplemental Figure 2). In adjusted Cox proportional hazards models, the effect of

PDR remained significant only for those on EFV-based ART (AHR 0.58, 95%CI 0.35, 0.96), but not for those DTG-based ART (AHR 1.01, 95%CI 0.80, 1.27).

Discussion:

In contrast to the effect seen with long-term outcomes, we did not find that PDR had an impact on time to initial suppression or change in quantified viral load from enrolment to 12 weeks, suggesting that NNRTI PDR affects longer term maintenance of suppression for DTG-based ART or via a non-virally mediated behavioral mechanism.

3. Confounding #1. A factor to be a confounder had to be a predictor of the outcome but also somewhat at least weakly causing pre-treated drug resistance. Most of the variables included in the logistic regression model do not satisfy this definition. In fact, the only important confounder is baseline viral load. Although including predictors of outcome should increase efficiency, it might be beneficial to have a more parsimonious model to avoid overfitting as confounding is unlikely to be introduced by the selected factors. Indeed, Supplemental Table 5 shows that some of these factors (i.e. sex age, education etc.) are quite weakly associated with the outcome. In addition, the model is adjusted for current measures of adherence such as pill count-based adherence (calculated as the number of pills taken since the prior visit and self-reported adherence (dichotomized as perfect adherence in the past 4 days). These are clearly mediators, not confounding factors as pre-existing resistant viruses become dominant via poor adherence after baseline. This is particularly true for EFV which is known to have a lower barrier to resistance than DTG so controlling for adherence would have removed an important part of the total effect. Personally I would only control for baseline viral load and NRTI resistance (both Sanger and 2-20% threshold in separate models) and use these models to test for the interaction.

Response

We agree with the reviewer that a simpler model might enable a clearer picture of the impact of PDR on outcomes by treatment arm, and be less susceptible to potential over-fitting and/or mediating effects of adherence. As such, we have updated our primary regression models and Table 3 to include three sets of models: 1) univariable models; 2) multivariable models adjusted only for baseline viral load, regimen and PDR (as suggested by the review) and 3) the previously described multivariable model including potential confounders and other predictors of treatment success.

In a third model intended to focus more directly on the virologic factors that determine treatment outcome, we restricted the model to pre-treatment viral load, presence or absence of PDR, and study treatment allocation.

We found similar effect sizes of PDR on treatment outcomes in all three models, suggesting minimal confounding effects:

Table 3. Logistic regression models for 96-week virologic success in the ADVANCE Trial^a

Covariable	Univariable Models		Baseline Viral Load-Adjusted Model		Fully Adjusted Multivariable Model	
	Odds Ratio	P-	Adjusted Odds	P-	Adjusted Odds	P-

	(95%CI)	value	Ratio (95%CI)	value	Ratio (95%CI)	value
Female Sex	0.90 (0.62, 1.29)	0.59			0.82 (0.54, 1.25)	0.35
Age (each year)	1.05 (1.02, 1.07)	<0.001			1.02 (0.99, 1.05)	0.14
Married or Partner	1.38 (0.86, 2.23)	0.18			0.95 (0.56, 1.60)	0.84
Tertiary education	0.83 (0.45, 1.54)	0.66			0.81 (0.41, 1.57)	0.53
Employed	2.07 (1.43, 2.98)	<0.001			1.77 (1.17, 2.67)	0.01
Pretreatment CD4 count						
≤200 cells/ μ L	REF				REF	
201-350 cells/ μ L	1.31 (0.83, 2.07)	0.25			1.27 (0.76, 2.13)	0.37
351-500 cells/ μ L	1.12 (0.67, 1.87)	0.66			0.98 (0.54, 1.77)	0.95
>500 cells/ μ L	1.13 (0.68, 1.88)	0.63			0.99 (0.54, 1.83)	0.97
Pre-treatment viral load						
<10,000 copies/mL	REF		REF		REF	
10,000-100,000 copies/mL	0.59 (0.37, 0.92)	0.02	0.56 (0.36, 0.89)	0.01	0.52 (0.31, 0.88)	0.01
>100,000 copies/mL	0.49 (0.29, 0.82)	0.006	0.51 (0.30, 0.85)	0.01	0.39 (0.21, 0.72)	0.003
Low self-reported adherence ^b (n, %)	0.36 (0.25, 0.52)	<0.001			0.41 (0.27, 0.63)	<0.001
Pill count adherence (n, %) ^c						
<90%	REF				REF	
90-95%	2.99 (1.39, 6.43)	0.005			2.71 (1.15, 6.38)	0.02
95-100%	6.16 (3.31, 11.46)	<0.001			3.51 (1.70, 7.24)	0.001
Regimen						
Efavirenz-based regimen	REF		REF		REF	
Dolutegravir-based regimen	0.80 (0.54, 1.18)	0.26	0.90 (0.22, 0.53)		1.02 (0.67, 1.57)	0.92
Presence of WHO-defined pretreatment drug resistance	0.33 (0.22, 0.52)	<0.001	0.34 (0.22, 0.53)	<0.001	0.38 (0.23, 0.61)	<0.001

Because our primary exposure of interest in this study, namely pre-treatment drug resistance, was not randomly allocated, and because we are fortunate to have a reasonably large sample size to fit models with known correlates (based on published data) of treatment success, we feel the optimal approach for model efficiency to estimates correlates of virologic suppression and to consider potential complex confounding relationships is to present adjusted models inclusive of all correlates of virologic suppression.

4. Confounding #2. Authors are clearly aware of the possibility of unmeasured confounding and I was impressed by the calculation of the E-value. However, INSTI resistance (both low level and >20% level) is clearly an unmeasured confounder that could explain the high rate of failure in the DTG arm. For

example, I have anecdotal evidence that G140ACRS in INSTI region even if detected at 2-20% level can increase the risk of failing INSTI-based therapy by 60%.

Response

Unfortunately, our pre-treatment deep sequencing did not include the INSTI gene. We think it unlikely that major INSTI resistance is a confounder of the relationship between PDR and treatment outcomes in this study. As the reviewer knows, there was essentially 0% INSTI use in this population at the time of the study, and numerous studies have investigated for the presence of INSTI resistance in South Africa and generally noted mutations that confer resistance to dolutegravir present in <1% of the population.¹⁴⁻¹⁶ That said, we agree that low level resistance and/or as of yet undetermined viral properties related to NNRTI resistance that affect INSTI efficacy remains a strong possibility:

Finally, our sequencing did not include the integrase region of the pol gene. Although resistance mutations that confer resistance to dolutegravir remain rare in South Africa, we cannot exclude the possibility of low-level resistance as a possible contributor to our findings.^{17,18}

Furthermore, there is confounding by NRTI resistance that has not been accounted for in the analysis. Figure 2 clearly shows that both M184V and K65R were more prevalent in people starting DTG.

Response

M184V (1.4% prevalence) and K65R (0.9% prevalence) were extremely rare in this cohort and thus are very unlikely to be cause of our findings. Indeed only, 2.3% of individuals had any NRTI resistance mutations at enrollment. To explore the possibility of NRTI resistance specifically as a contributor to our findings of the impact on DTG suppression, we compared the contributions of any PDR (including NRTI and/or NNRTI) with that of K103N resistance alone, and see no major differences in these in the dolutegravir arms:

	Total Cohort			Efavirenz arm			Dolutegravir arms			Interaction P-value ^a
	PDR	No PDR	P-value	PDR	No PDR	P-value	PDR	No PDR	P-value	
Primary Outcome (any NRTI or NNRTI mutation)	73/112 (65%)	606/714 (85%)	<0.001	12/20 (60%)	215/249 (86%)	0.002	61/92 (66%)	391/465 (84%)	<0.001	0.39
Primary Outcome (K103N only)	34/49 (69%)	611/724 (84%)	0.006	7/8 (88%)	215/251 (86%)	0.88	27/41 (66%)	396/473 (84%)	0.004	0.35
Mean change in log ₁₀ viral load from baseline to 12 weeks (any NRTI or NNRTI mutation)	2.63 (0.95)	2.66 (0.79)	0.78	2.00 (0.73)	2.61 (0.75)	0.004	2.80 (0.94)	2.68 (0.81)	0.34	0.001
log ₁₀ viral load from baseline to 12 weeks (K103N only)	2.67 (0.92)	2.65 (0.80)	0.88	2.14 (0.51)	2.59 (0.77)	0.08	2.79 (0.95)	2.68 (0.82)	0.44	0.33

M184I should be also counted in as it confers the same reduction in susceptibility of FTC as M184V. Again, M184VI at 2-20% can be shown to be associated with 3-fold increase in risk of virological failure of triple combination including 3TC/FTC plus an anchor drug including INSTI, a value which exceeds the calculated E-value making residual confounding a possible explanation for the findings.

Response

This was our mistake – we included both M184V and I in our characterization of M184 mutations and have updated the figure to reflect this:

*NRTI: nucleos(t)ide reverse transcriptase inhibitor;
NNRTI: non-nucleoside reverse transcriptase inhibitor

5. Interaction. The analysis was clearly not powered to test this interaction so a p-value<0.05 for the test does not mean that interaction can be ruled out. Looking at some of the analyses shown in Table 2 the risk difference by PDR in EFV are much larger (almost a 3-fold difference) than those in DTG. For example for the secondary outcome (93%-68% = 25%) for EVF vs. (94%-85%=9%) for DTG and similarly for the 48-week snapshot (81%-46%=35%) for EFV vs. (86%-74%=12) for DTG. On the basis of these differences and all the caveats described above I do not think that the data support that the association between PDR and risk of failure does not vary by treatment arm.

Response

We agree with this point and have toned down the language about differences by arm throughout the manuscript.

6. Lines 242-243. NNRTI resistance detected at 2-20% of virus population has been consistently shown to be associated with failure of EFV-based regimens. [Casadellà M, Paredes R. Deep sequencing for HIV-1 clinical management. Virus Res. 2017 Jul 15;239:69-81]. The fact that these results could not be replicated here, shades doubts on the credibility of the rest of the findings.

Response

While we agree with the reviewer that multiple studies have demonstrated relationships between resistance at minority (<20%) frequencies and treatment outcomes, there have also been multiple studies that have shown the opposite. For example, a recent systematic review of 25 studies that have

investigated the impact of low-abundance frequency mutations and virologic outcomes found that 56% of studies did not demonstrate a significant relationship between the two.¹⁹ Notably, all studies in that review from South Africa in particular failed to demonstrate relationships between minority frequency resistance variants and virologic suppression. Thus, while there is some evidence in support of this relationship, we do not agree that available data have fundamentally proven this relationship exists. To more specifically explore this relationship in our cohort in response to the reviewer's concerns, we have modified our analysis to focus on minority frequency (2-20%) specifically impacting NNRTI regimens, as that has been the focus of prior studies which have done so:

Supplemental Table 4. Virologic success in the ADVANCE Trial by the presence of WHO-defined pretreatment minority drug resistance as defined by a mutation frequency 2-20%^a

	Total Cohort			Efavirenz arm			Dolutegravir arms			Interaction P-value
	PDR	No PDR	P-value	PDR	No PDR	P-value	PDR	No PDR	P-value	
Primary Outcome	32/36 (89%)	575/680 (85%)	0.48	8/10 (80%)	207/239 (87%)	0.55	24/26 (92%)	368/441 (83%)	0.23	0.22
Secondary Outcome	34/35 (97%)	614/654 (97%)	0.43	9/10 (90%)	209/224 (93%)	0.69	25/25 (100%)	405/430 (94%)	0.22	N/A ^f
48-week Snapshot	32/41 (78%)	600/713 (84%)	0.30	10/10 (100%)	203/255 (80%)	0.11	22/31 (71%)	397/458 (87%)	0.02	N/A ^f
96-week Snapshot	30/41 (73%)	565/713 (79%)	0.35	10/10 (100%)	188/255 (74%)	0.06	20/31 (65%)	377/458 (82%)	0.01	N/A ^f
Mean change in log ₁₀ viral load from baseline to 12 weeks (SD)	2.65 (0.78)	2.66 (0.79)	0.96	2.41 (0.73)	2.62 (0.75)	<0.001	2.75 (0.78)	2.68 (0.62)	0.69	0.53

Other points

1. Lines 197-199. At the time of data extraction, all 873 had completed observation up to 96 weeks. A total of 48 and 91 individuals were excluded from the primary and secondary analyses, respectively, for not remaining in the study to 12 or 24 weeks. These two sentences seem to contradict each other?

Response

Thank you for noting this error we have corrected the text and Figure 1 with the correct figures after updating definitions of outcomes and inclusion criteria based on reviewer feedback:

Of participants included in PDR analyses, 289 (33%) were randomized to an EFV-based regimen and 585 (67%) were randomized to a DTG-based regimen. At the time of data extraction, all had completed observation up to 96 weeks. A total of 48 and 82 individuals were excluded from our primary and secondary analyses, respectively, for not remaining in the study to 12 or 24 weeks.

2. Line 207. Typo.individual had a nucleoside, not 'an'

Response

Thank you – we have corrected this.

3. Lines 221 and 320. Was the E-value equal to 2.9 or 2.8 ? Which exact analysis was used to calculate this (primary endpoint?)

Response

Thank you for finding that inconsistency. We use our multivariable logistic regression model adjusted for known confounders to derive this (displayed in Table 3). After updating our definitions based on reviewer responses, the odd's ratio was 0.38 (95%CI 0.23, 0.62), corresponding to an updated E-value of 2.7. We have updated the manuscript throughout with this.

4. Line 248. We report a strong and pervasive association between NNRTI resistance...Although the majority of participants harboured NNRTI resistance the exposure was defined as having ≥ 1 mutations in the WHO list.

Response

As mentioned in the prior response, the vast majority of individuals had only NNRTI resistance (<3% with others), as such we focus on NNRTI resistance, which differentiates this cohort meaningfully from others with multi-class resistance at the time of first-line failure. To more accurately describe this cohort, we have updated that sentence as follows:

We report a strong association between drug resistance before treatment initiation, primarily to the NNRTI class, and virologic failure for people initiating first-line ART in the ADVANCE clinical trial.

5. Lines 256-258. The authors use the paper by Clutter DS et al in support of their hypothesis that NNRTI resistance might impair response to INSTI-based regimens. The argument seems very speculative especially because the paper actually supports the opposite (INSTI-based regimens are an effective option with at least equal efficacy compared with bPIs for patients with isolated NNRTI TDR).

Response:

We agree this is a hypothesis-generating finding only. We have updated the reference with a more appropriate paper in support of this (Hu et al) and have toned down that statement to be more cautionary:

The observed effect we identified may be consistent with preliminary data suggesting higher replication of NNRTI mutant viruses in the context of drug pressure from integrase inhibitors,²⁸

6. Lines 266-268. Current adherence is a mediator, not a confounder. Although there is suspicion that some may not be ART-naïve patients, clearly the temporality goes against believing that these measures could be confounders.

Response

While we agree with the reviewer that poor adherence could be an effect modifier, because it would differentially affect suppression in those with and without PDR, in this case we suspect that adherence behaviors might also serve as a confounder of the relationship between PDR and

treatment failure. If PDR is a surrogate marker for prior treatment default and poor adherence then those with PDR might appear to have worse outcomes due to poor adherence rather than the presence of PDR itself. Importantly, the effect size of PDR on our outcomes did not change meaningfully after addition of adherence measures to our model. To better demonstrate that adherence and PDR appeared to have additive effects (but not confounding or interactive effects), we have added a figure to demonstrate how both PDR and treatment adherence impacted virologic suppression in both arms (Figure 4):

Figure 4. 96-week treatment outcomes among participants in the ADVANCE Trial divided by treatment arm, presence or absence of WHO-defined pre-treatment drug resistance, and achievement of greater than versus less than 95% adherence based on pharmacy pill count.

*Drug resistance defined by presence of World Health Organization-defined Drug Resistance Mutations to Nucleoside or Non-Nucleoside Reverse Transcriptase Inhibitors prior to ART Initiation

7. Line 276. Typo. DTG based ART has, not 'in has'

Response

Thank you – we have corrected this.

8. Line 307. ...between 5-20% of viral quasispecies. In the Results section it says 2-20%, which one is correct? The choice would impact on the prevalence of exposure and therefore on the results. Indeed, it is possible that statistical power is lower for the 5-20% threshold vs. 2-20% threshold.

Response

Thank you for noting this error. We have updated the manuscript and analysis to focus on the 2-20% range of minority variants.

9. Lines 314-315. ...because study arm was determined by computer randomisation. I do not understand this. Exchangeability created by randomisation was broken down by the fact that plasma samples were not available in all participants.

Response

The rates of inclusion (or not) in this analysis was balanced evenly across study arms. Availability of plasma samples, which were collected at baseline on the day of randomization, or failure to sequence them should have been similar between randomized study arms. Thus, any imbalance in resistance by treatment arm could only have happened by chance. We feel that this is an important limitation that remains pertinent to the manuscript.

10. Table 2. I would remove the p-values within the strata as these are subset analyses so difficult to interpret.

In comment 3 above, the reviewer requests a simplified model to test for interaction, which we have done for Tables 2, Supplemental Tables 3 and 4. That said, we would be happy to remove these P-values from the tables if advised by the reviewer or editor. Notably the addition of the virologic change from baseline to 12 weeks as also suggested by the reviewer does demonstrate an interactive effect by treatment group.

Reviewer #3 (Remarks to the Author):

The authors assess the efficacy of HIV-1 integrase inhibitors in patients with drug resistance mutations in reverse transcriptase using samples from the ADVANCE trial. Some comments regarding the statistical analyses:

1. The definition of virologic success is a little confusing. If someone achieved a viral load (VL) of <1000 but ≥ 200 from 12 weeks onward considered a success for the entire time period? I understand that the authors would like to simplify the outcome so there is only one outcome per person, but if this participant then had a VL ≥ 200 at 24 weeks were they still considered a success?

Response

We apologize for this confusion and agree we should have made our primary outcome definition clearer. To answer the reviewer's specific inquiry, if a participant has a VL ≥ 200 at any time after 12 weeks they were considered a failure – so in the example posed (with a VL over 200 at 24 weeks) they would be considered a failure. We have reworded the definition to add clarity:

Our primary outcome of interest was 96-week virologic success, which we defined as a sustained a viral load <1000 copies/mL from 12 weeks through 96weeks, <200 copies/mL from 24 weeks through 96 weeks, and <50 copies/mL from 48 weeks through 96 weeks.

2. With the longitudinal nature of the data one definition of viral suppression could be used and a time to event analysis done to look at time to viral suppression. Did the authors consider this?

Response

We appreciate this recommendation and have added a survival analysis to our methods and results. This is now discussed in the methods, results and discussions sections and we have added Supplemental Figure 2 to demonstrate the K-M curves generated from this analysis:

3. With the outcome (viral suppression) being very common, log binomial regression should probably be used instead of logistic regression to give risk ratios instead of odds ratios.

Response

In response to the reviewer comments we explored use of binomial logistic regression, but we had difficulty getting our multivariable models to converge (a common occurrence for this type of model). That said, we agree that odds ratios can be misinterpreted. To enhance interpretation of our results, we provide multiple examples of crude results in Figure 3 (crude rates of success by outcome, arm and PDR), Supplemental Figure 2 (newly added K-M plots), as well as in Table 2, and Supplemental Tables 3 and 4. If the reviewer or editors feel strongly about a risk ratio model, we would be happy to explore Poisson regression models with robust standard errors or converting our multivariable logistic regression models to proportional rates using post-regression marginal techniques.

4. The authors assessed if there were differences in clinical or demographic characteristics between those who successfully underwent sequencing and those who did not. The same should be done to compare the 48 and 91 individuals who did not remain in the study for 12 or 24 weeks.

Response:

We have updated Supplemental Table 1 to include three categories of individuals: 1) those in the primary analysis; 2) those in the 96-snapshot analysis but not in the primary analysis; 3) those not in any analysis. Perhaps not surprisingly, we found that those who dropped out of the study before 96-weeks were more likely than the other two groups to be unemployed. Otherwise the characteristics were largely similar between groups.

Supplemental Table 1. Clinical and demographic variables comparing individuals included and excluded in the ADVANCE Trial pretreatment drug resistance analytic sample

	Included in Primary Outcome Analysis (n=826)	Included in Snapshot Analyses but not Primary Outcome (n=48)	Excluded from Analytic Dataset (n=179)	P-value ^a
Female sex (n, %)	494 (59.8%)	27 (56.3%)	102 (57.0%)	<0.001
Age (median, IQR)	32 (27-38)	30 (25-33)	31 (26-37)	0.04
Married or Partner (n, %)	168 (20.4%)	10 (20.8%)	34 (19.0%)	0.91
Tertiary education (n, %)	69 (8.4%)	8 (16.7%)	14 (7.8%)	0.13
Employed (n, %)	519 (63.8%)	20 (41.7%)	102 (58.0%)	0.005
Pretreatment CD4 count (n, %)				0.83
≤200 cells/uL	259 (31.4%)	12 (25.0%)	51 (28.5%)	
201-350 cells/uL	247 (30.0%)	15 (31.3%)	49 (27.4%)	
351-500 cells/uL	157 (19.0%)	11 (22.9%)	37 (20.7%)	
>500 cells/uL	163 (19.7%)	10 (20.8%)	42 (23.5%)	
Pretreatment viral load (n, %)				0.04
<10,000 copies/mL	260 (31.5%)	24 (50.0%)	61 (34.1%)	
10,000-100,000 copies/mL	377 (45.6%)	20 (41.7%)	83 (46.4%)	
>100,000 copies/mL	189 (22.9%)	4 (8.3%)	35 (19.6%)	
Study arm (n, %)				0.45
DTG/TAF/FTC	272 (32.9%)	17 (35.4%)	62 (34.6%)	
DTG/TDF/FTC	285 (34.5%)	11 (22.9%)	55 (30.7%)	
EFV/TDF/FTC	269 (32.6%)	20 (41.7%)	62 (34.6%)	
Presence of WHO-defined pretreatment drug resistance (n, %)	112 (86.4%)	10 (20.8%)	11 (11.5%)	0.30
Pill count ^{b,c} (n, %)				0.75
<90%	45 (5.5%)	3 (10.3%)	9 (5.1%)	
90-95%	81 (9.9%)	2 (6.9%)	20 (11.4%)	
>95%	696 (84.7%)	24 (82.8%)	146 (83.4%)	

5. Table 1 compares characteristics by regimen for those with successful sequencing. Given the primary analysis does not include the 48 individuals who did not remain in the study for 12 weeks, it seems that these individuals should be excluded from this table.

Response

We have updated Table 1 to compare those included only the primary outcome analysis.

Table 1. Cohort characteristics for participants who completed pre-treatment HIV drug resistance testing and included in our primary analysis of virologic failure, divided by regimen

	Efavirenz arm (n=269)	Dolutegravir arms (n=557)	P-value ^a
Female sex (n, %)	153 (56.9%)	341 (61.2%)	0.23
Age (median, IQR)	32 (27-37)	32 (27-38)	0.83
Married or Partner (n, %)	60 (22.3%)	108 (19.4%)	0.34
Tertiary education (n, %)	18 (6.7%)	51 (9.2%)	0.22
Employed (n, %)	170 (63.7%)	349 (63.8%)	0.97
Pretreatment CD4 count (n, %)			0.58
≤200 cells/uL	80 (29.7%)	179 (32.1%)	
201-350 cells/uL	81 (30.1%)	166 (29.8%)	
351-500 cells/uL	58 (21.6%)	99 (17.8%)	
>500 cells/uL	50 (18.6%)	118 (20.3%)	
Pretreatment viral load (n, %)			0.33
<10,000 copies/mL	89 (33.1%)	171 (30.7%)	
10,000-100,000 copies/mL	113 (42.0%)	264 (47.4%)	
>100,000 copies/mL	67 (24.9%)	122 (21.9%)	
Low self-reported adherence ^b (n, %)	113 (42.0%)	252 (45.2%)	0.38
Pill count adherence (n, %) ^c			0.45
<90%	12 (4.5%)	33 (6.3%)	
90-95%	23 (8.6%)	58 (10.5%)	
95-100%	233 (87.0%)	463 (83.6%)	
Presence of Any WHO-defined pretreatment drug resistance	20 (7.4%)	92 (16.5%)	<0.001

6. Table 3: There are a lot of variables in the multivariable model and some of these are likely to be collinear (e.g., low self reported adherence and pill count adherence). Please assess collinearity of all variables and adjust the analyses accordingly.

Response:

We thank the reviewer for this recommendation. We have assessed collinearity for our primary regression model and found little evidence of such added these results in Supplemental Table 7:

Supplemental Table 7. Assessment of collinearity between included covariates in our multivariable regression model of our primary outcome

Variable	Variation Initiation Factor (VIF)	1/VIF
Female Sex	1.1	0.908709
Age (each year)	1.14	0.877584
Married or Partner	1.08	0.92839
Tertiary education	1.02	0.98108
Employed	1.11	0.903705
Pretreatment CD4 count		
≤200 cells/uL	REF	--

201-350 cells/uL	1.45	0.690362
351-500 cells/uL	1.49	0.67171
>500 cells/uL	1.62	0.618711
Pre-treatment viral load		
<10,000 copies/mL	REF	--
10,000-100,000 copies/mL	1.49	0.671991
>100,000 copies/mL	1.66	0.603334
Low self-reported adherence	1.1	0.911344
Pill count adherence		
<90%	REF	--
90-95%	2.71	0.368918
95-100%	2.85	0.350324
Efavirenz vs dolutegravir	1.03	0.972675
Presence of WHO-defined pretreatment drug resistance	1.04	0.959658

Response To Review References Cited

- 1 Shi, L. *et al.* Concordance of Adherence Measurement Using Self-Reported Adherence Questionnaires and Medication Monitoring Devices. *Pharmacoeconomics* **28**, 1097-1107, doi:10.2165/11537400-000000000-00000 (2010).
- 2 Musinguzi, N. *et al.* Comparison of subjective and objective adherence measures for preexposure prophylaxis against HIV infection among serodiscordant couples in East Africa. *Aids* **30**, 1121-1129, doi:10.1097/qad.0000000000001024 (2016).
- 3 Okatch, H. *et al.* Brief Report. *JAIDS Journal of Acquired Immune Deficiency Syndromes* **72**, 542-545, doi:10.1097/qai.0000000000000994 (2016).
- 4 Hamers, R. L. *et al.* Patterns of HIV-1 drug resistance after first-line antiretroviral therapy (ART) failure in 6 sub-Saharan African countries: implications for second-line ART strategies. *Clin Infect Dis* **54**, 1660-1669, doi:10.1093/cid/cis254 (2012).
- 5 Inzaule, S. C. *et al.* Previous antiretroviral drug use compromises standard first-line HIV therapy and is mediated through drug-resistance. *Scientific Reports* **8**, doi:10.1038/s41598-018-33538-0 (2018).
- 6 Manne-Goehler, J. *et al.* ART Denial: Results of a Home-Based Study to Validate Self-reported Antiretroviral Use in Rural South Africa. *AIDS and behavior* **23**, 2072-2078, doi:10.1007/s10461-018-2351-7 (2018).
- 7 Kim, A. A. *et al.* Undisclosed HIV infection and antiretroviral therapy use in the Kenya AIDS indicator survey 2012. *Aids* **30**, 2685-2695, doi:10.1097/qad.0000000000001227 (2016).
- 8 Grabowski, M. K. *et al.* The validity of self-reported antiretroviral use in persons living with HIV. *Aids*, doi:10.1097/qad.0000000000001706 (2017).
- 9 VanderWeele, T. J. & Ding, P. Sensitivity Analysis in Observational Research: Introducing the E-Value. *Annals of internal medicine* **167**, doi:10.7326/m16-2607 (2017).
- 10 Hamers, R. L. *et al.* Effect of pretreatment HIV-1 drug resistance on immunological, virological, and drug-resistance outcomes of first-line antiretroviral treatment in sub-Saharan Africa: a multicentre cohort study. *The Lancet Infectious Diseases* **12**, 307-317, doi:10.1016/s1473-3099(11)70255-9 (2012).
- 11 Panpradist, N. *et al.* OLA-Simple: A software-guided HIV-1 drug resistance test for low-resource laboratories. *EBioMedicine* **50**, 34-44, doi:10.1016/j.ebiom.2019.11.002 (2019).
- 12 Chung, M. H. *et al.* Evaluation of the management of pretreatment HIV drug resistance by oligonucleotide ligation assay: a randomised controlled trial. *Lancet HIV* **7**, e104-e112, doi:10.1016/S2352-3018(19)30337-6 (2020).
- 13 Dolgin, E. Long-acting HIV drugs advanced to overcome adherence challenge. *Nature Medicine* **20**, 323-324, doi:10.1038/nm0414-323 (2014).
- 14 Brado, D. *et al.* Analyses of HIV-1 integrase sequences prior to South African national HIV-treatment program and availability of integrase inhibitors in Cape Town, South Africa. *Scientific Reports* **8**, doi:10.1038/s41598-018-22914-5 (2018).
- 15 Obasa, A. E. *et al.* Drug Resistance Mutations Against Protease, Reverse Transcriptase and Integrase Inhibitors in People Living With HIV-1 Receiving Boosted Protease Inhibitors in South Africa. *Frontiers in Microbiology* **11**, doi:10.3389/fmicb.2020.00438 (2020).

- 16 Fish, M. Q. *et al.* Natural Polymorphisms of integrase Among HIV Type 1-Infected South African Patients. *AIDS research and human retroviruses* **26**, 489-493, doi:10.1089/aid.2009.0249 (2010).
- 17 Inzaule, S. C. *et al.* Primary resistance to integrase strand transfer inhibitors in patients infected with diverse HIV-1 subtypes in sub-Saharan Africa. *Journal of Antimicrobial Chemotherapy* **73**, 1167-1172, doi:10.1093/jac/dky005 (2018).
- 18 Derache, A. *et al.* Predicted antiviral activity of tenofovir versus abacavir in combination with a cytosine analogue and the integrase inhibitor dolutegravir in HIV-1-infected South African patients initiating or failing first-line ART. *Journal of Antimicrobial Chemotherapy* **74**, 473-479, doi:10.1093/jac/dky428 (2019).
- 19 Metzner, K. J. *et al.* Low-Abundance Drug-Resistant HIV-1 Variants in Antiretroviral Drug-Naive Individuals: A Systematic Review of Detection Methods, Prevalence, and Clinical Impact. *The Journal of Infectious Diseases* **221**, 1584-1597, doi:10.1093/infdis/jiz650 (2020).
- 20 Clutter, D. S. *et al.* Response to Therapy in Antiretroviral Therapy-Naive Patients With Isolated Nonnucleoside Reverse Transcriptase Inhibitor-Associated Transmitted Drug Resistance. *J Acquir Immune Defic Syndr* **72**, 171-176, doi:10.1097/QAI.0000000000000942 (2016).

REVIEWERS' COMMENTS

Reviewer #1 (Remarks to the Author):

I do thank the authors for addressing the comments, I have no additional comments.

Reviewer #2 (Remarks to the Author):

The authors did a great job at revising the manuscript and partially lowering the tone regarding the effect of NNRTI resistance for people receiving DTG-based regimens. Indeed, the new analyses, especially, when focussing on virological potency carry little evidence for such an association.

I have only a couple of additional minor points.

First, regarding the review cited as Ref #50. Although the paper is interesting it is not a formal meta-analysis and little effort has been made in matching intervention for the actual regimen used. Thus, for example, almost all studies showing no association between low level NNRTI variants and virological response are new studies involving Rilpivirine and not Efavirenz and we know that the genetic barrier of Rilpivirine is much higher. I think that this should be noted when Discussing the results.

Second, regarding my original last minor point I was only suggesting to remove the p-values of the analysis within the strata and keep only the interaction p-values.

RESPONSE TO REVIEWERS AND EDITORIAL COMMENTS

REVIEWERS' COMMENTS SECOND REVIEW

Reviewer #1 (Remarks to the Author):

I do thank the authors for addressing the comments, I have no additional comments.

Response

We thank the reviewer for their time and thorough repeated review of this work.

Reviewer #2 (Remarks to the Author):

The authors did a great job at revising the manuscript and partially lowering the tone regarding the effect of NNRTI resistance for people receiving DTG-based regimens. Indeed, the new analyses, especially, when focussing on virological potency carry little evidence for such an association. I have only a couple of additional minor points.

Response

We thank the reviewer for their time and thorough repeated review of this work.

First, regarding the review cited as Ref #50. Although the paper is interesting it is not a formal meta-analysis and little effort has been made in matching intervention for the actual regimen used. Thus, for example, almost all studies showing no association between low level NNRTI variants and virological response are new studies involving Rilpivirine and not Efavirenz and we know that the genetic barrier of Rilpivirine is much higher. I think that this should be noted when Discussing the results.

Response

We agree that studies of minority resistance have shown conflicting results, and we have taken pains to ensure that we avoid conclusive statements on one side or the other. To respond to the reviewer, we have modified the phrase to include that the studies of newer regimens are less likely to have this relationship:

*However, many studies, and particularly those considering newer ART regimens, have failed to demonstrate a role for low-level mutant viruses in determining clinical outcomes.*⁵³

Second, regarding my original last minor point (*Table 2. I would remove the p-values within the strata as these are subset analyses so difficult to interpret*) I was only suggesting to remove the p-values of the analysis within the strata and keep only the interaction p-values.

Response

We agree that stratified analyses can be fraught, particularly with type II error issues. However, in this case the stratified analyses by treatment arm in Table 2 are highly statistically significant – suggesting no power issue, with the notable exception of the virologic outcome for dolutegravir that the reviewer thoughtfully request we add, and shows that the virologic outcome was *not* affected by NNRTI resistance in that arm. Thus – we feel these do add context to our results and would prefer to maintain these unless the reviewer or editors feel otherwise.

Table 2. Virologic success in the ADVANCE Trial by the presence of WHO-defined pretreatment drug resistance

	Total Cohort			Efavirenz arm			Dolutegravir arms			Interaction
	PDR	No PDR	P-value	PDR	No PDR	P-value	PDR	No PDR	P-value	P-value ^a
Primary Outcome ^b	73/112 (65%)	606/714 (85%)	<0.001	12/20 (60%)	215/249 (86%)	0.002	61/92 (66%)	391/465 (84%)	<0.001	0.39
Secondary Outcome ^c	86/105 (82%)	646/687 (94%)	<0.001	13/19 (68%)	218/234 (93%)	<0.001	73/86 (85%)	428/453 (94%)	0.001	0.26
48-week Snapshot ^d	84/122 (69%)	630/752 (84%)	<0.001	11/24 (46%)	213/265 (80%)	<0.001	73/98 (75%)	417/487 (86%)	0.006	0.10
96-week Snapshot ^d	71/122	593/752	<0.001	11/24 (46%)	198/265	0.002	60/98 (61%)	395/487	<0.001	0.64

Mean change in log ₁₀ viral load from baseline to 12 weeks (SD)	(58%)	(79%)		(75%)			(81%)			
	2.63 (0.95)	2.66 (0.79)	0.78	2.00 (0.73)	2.61 (0.75)	0.004	2.80 (0.94)	2.68 (0.81)	0.34	0.001